# Effect of Hemp Protein and Sea Buckthorn Extract on Quality and Shelf Life of Cooked-Smoked Sausages

**DOI:** 10.3390/foods14152730

**Published:** 2025-08-05

**Authors:** Kainar Bukarbayev, Sholpan Abzhanova, Lyazzat Baibolova, Gulshat Zhaksylykova, Talgat Kulazhanov, Vitalii Vasilenko, Bagila Jetpisbayeva, Alma Katasheva, Sultan Sabraly, Yerkin Yerzhigitov

**Affiliations:** 1Department of Food Biotechnology, Almaty Technological University, Almaty 050000, Kazakhstan; 2Department of Machines and Equipment for Food Production, Voronezh State University of Engineering Technologies, Voronezh 394036, Russia; 3Department of Energy Saving and Automation, Kazakh National Agrarian Research University, Almaty 050010, Kazakhstan

**Keywords:** hemp protein, cooked-smoked sausage, sea buckthorn extract, technological scheme, extrudate, storage

## Abstract

Modern meat processing faces several challenges, including high resource consumption, environmental impact, and the need to enhance the nutritional and biological value of finished products. In this context, interest is growing in functional plant-based ingredients capable of improving the quality of meat products. The aim of this study was to evaluate the effect of adding 0.01% hemp protein powder and 0.01% sea buckthorn extract (based on the weight of unsalted raw material) on the nutritional, technological, and microbiological characteristics of cooked-smoked sausages. The results demonstrated an increase in total protein content, a 2.5-fold rise in tocopherol levels, as well as a 17.9% improvement in the Amino Acid Score of threonine and a 2.48% increase in the biological value of protein. Samples enriched with plant-based components exhibited enhanced organoleptic properties and greater storage stability over 36 days. In addition, extrusion parameters for the production of the protein additive were optimized, resulting in a stable functional ingredient.

## 1. Introduction

At the current stage of development in the food industry, particular attention is being paid to creating functional meat products with enhanced properties. This is driven by the need to improve nutritional and biological value and product quality and shelf life, while ensuring compliance with new environmental sustainability and healthy eating standards. The shift towards using natural ingredients and reducing the amount of animal protein in formulations is influenced by increasing consumer demands and trends towards sustainable industry development [1].

In this regard, plant-based techno-functional components, such as protein isolates and natural extracts with pronounced antioxidant and antimicrobial properties, are of particular interest. The use of these components not only optimises product formulations and technological processes but also enables the development of products with enhanced sensory qualities and improved consumer acceptance [2,3].

Like most meat products, sausages are a valuable source of complete protein [4]. Therefore, when improving sausage formulations, it is essential to consider the content and bioavailability of polyunsaturated fatty acids, proteins, macronutrients, and micronutrients, as well as vitamins, in the final product. The established perception of meat products among consumers encourages manufacturers to seek ways to enhance their health benefits and develop healthier, more innovative meat products.

In recent years, nutritionists and food technologists have made considerable efforts to develop novel meat products enriched with natural antioxidants, antimicrobial compounds, dietary fibres, and alternative protein sources. There is a growing interest among consumers in products containing bioactive or functional components that may provide additional health benefits [5]. Plant-derived, non-meat ingredients are increasingly utilised as techno-functional agents, serving as binders, emulsifiers, fillers, or partial meat substitutes. Isolated plant proteins, in particular, are widely applied due to the rising cost of animal-based proteins and the favourable functional properties exhibited by many plant proteins.

One of the current challenges in the food industry is the replacement of protein isolates derived from genetically modified raw materials. Ongoing research focuses on identifying alternative sources of plant proteins from non-GMO crops. Over the past five years, various types of plant protein isolates that meet technological and functional requirements have emerged on the domestic market. These include isolates obtained from industrial hemp, wheat, flaxseed, and peas. Such proteins are characterised by high protein content, strong water-binding and emulsifying capacities, and relatively low production costs. For instance, unlike soy protein isolates, hemp, wheat, and pea proteins do not form gels but instead contribute directly to structure formation in meat systems. The potential for incorporating plant extracts and hemp protein into meat products is currently significant and continues to grow in response to market demand.

Sea buckthorn extract is one notable plant extract with a high concentration of antioxidants and other biologically active compounds. It enhances the nutritional value of food products, reduces oxidative reactions, and increases shelf life. Sea buckthorn (*Hippophae rhamnoides*) is a valuable and unique plant which has recently gained global attention due to its medicinal and nutritional potential [6]. All parts of the plant are considered to be rich sources of numerous biologically active substances. Sea buckthorn extract is characterised by a low protein content (0.7–3%) in fresh fruit but a high fat content (2.5–5%), particularly of unsaturated fatty acids. It also contains high levels of vitamin C (up to 400–700 mg/100 g of fresh fruit), carotenoids (50–100 mg/100 g of dry matter), and phenolic compounds [7].

Hemp protein is a plant-derived protein source that contains all the essential amino acids required by the human body. According to the literature, hemp protein powder contains approximately 40–52% protein, while higher-purity forms (isolates) may contain up to 90% protein. It primarily consists of globulins (edestin and albumin) and also includes dietary fibre (approximately 20%) and beneficial fats [8]. Hemp protein is also rich in antioxidants, vitamins, and minerals, making it a nutritious option for vegetarians, vegans, and individuals seeking to reduce their intake of animal-based proteins. Hemp (*Cannabis sativa* L.) produces highly nutritious seeds that have been consumed by humans for thousands of years [9]. At present, hemp seeds are mainly processed through mechanical pressing to extract valuable oil, while the resulting meal is utilised in the production of various protein-rich food products [10].

The incorporation of sea buckthorn extract and hemp protein into meat products contributes to a reduction in animal protein consumption, thereby promoting human health and reducing the environmental impact of meat production [11]. Advances in modern technologies and processing methods enable the efficient extraction and utilisation of these plant-based components while preserving their nutritional and functional properties [12]. Increasing consumer awareness of healthy lifestyles is also driving the demand for meat products formulated with plant-derived ingredients [13]. Overall, the application of sea buckthorn extract and hemp protein in meat products demonstrates considerable market potential and industrial relevance [14].

The scientific novelty and relevance of this study lie in investigating the enrichment of cooked-smoked meat products with functional, plant-based ingredients to improve their nutritional quality and microbiological stability.

This research primarily aims to evaluate the impact of the hemp protein and sea buckthorn extract on the quality of cooked-smoked sausages.

This study provides theoretical insights and practical implications for developing functional meat products. Incorporating plant-derived components such as hemp protein and sea buckthorn extract is a promising approach to innovating in the meat processing industry, particularly for producing sausages that are healthier and have a longer shelf life.

## 2. Materials and Methods

### 2.1. Materials

The following raw materials and reagents were used in this study: beef fillet, poultry fillet, and beef fat (Pervomayskiye Delicatessen LLP, Qoyankus, Kazakhstan); sodium nitrite (Sigma-Aldrich, Burlington, MA, USA); table salt (Aral as tұzy LLP, Kostanay, Kazakhstan); hemp protein powder (Zerde Scientific and Production Association LLP, Shymkent, Kazakhstan); sea buckthorn extract (Zerde Scientific and Production Association LLP, Shymkent, Kazakhstan); and granulated sugar, ground black pepper, nutmeg, basil, and mint (all purchased from local certified suppliers in Kazakhstan).

### 2.2. Extrusion of Hemp Protein Concentrate

Extrusion processing was carried out using a laboratory-scale twin-screw extruder (KDL-30, Munich, Germany) equipped with a die of 5 mm diameter. The raw material used was hemp protein concentrate with an initial moisture content of approximately 7%. Prior to extrusion, the concentrate was moistened by fine water spraying under continuous mixing to reach target moisture levels of 18%, 22%, and 26%. The amount of added water was calculated considering the initial moisture content. The moistened samples were equilibrated for 15 min to ensure uniform moisture distribution.

The experimental trials were designed according to a factorial plan with variations of the following parameters: initial moisture content of the raw material (x_1_): 18%, 22%, and 26%; screw rotation speed (x_2_): 0.9, 1.15, and 1.4 s^−1^; structural parameter (x_3_): 0.86 (dimensionless coefficient); and pressure in the pre-die zone (x_4_): 5, 7, and 9 MPa.

The extrusion process was carried out at a temperature range of 100–110 °C and monitored by built-in thermocouples located in the heating zones of the extruder. The pressure was measured using a digital pressure gauge (WIKA S-10, WIKA, Klingenberg, Germany). The screw rotation frequency was controlled by software with a precision of 0.01 s^−1^.

Moisture content of the samples before and after extrusion was determined using the gravimetric method in accordance with GOST 13586.5-93 [15]. Samples were dried to constant weight at 105 °C in a drying oven (Memmert UN-55, Schwabach, Germany), using analytical balances (Sartorius, accuracy 0.001 g, Munich, Germany).

#### 2.2.1. Determination of Moisture Content (Gravimetric Method)

The moisture contents of the hemp protein concentrate and extrudate samples were determined by the gravimetric method using a drying oven SNOL 30/130 (Umega Group, Ukmergė, Lithuania) and analytical balance Sartorius Practum 213 (Sartorius AG, Göttingen, Germany).

The method is based on drying the weighed sample to a constant mass, in accordance with GOST 13586.5-93, which aligns with international standards ISO 712:2009 [16] and AOAC 925.10 [17].

For analysis, a sample weighing 5.00 ± 0.01 g was placed in pre-dried and pre-weighed porcelain crucibles and then dried at 130 ± 2 °C for 40 min. After drying, the samples were cooled to room temperature (20–22 °C) in a desiccator (EXS-250, Glassco, Haryana, India) and reweighed.

The moisture content (W, %) was calculated using the following formula:W = ((*m*_1_ − *m*_2_)/*m*_1_) × 100(1)
where

*m*_1_—sample mass before drying (g);

*m*_2_—sample mass after drying (g).

Each sample was analysed in duplicate, and the result was reported as the mean value. The deviation between parallel determinations did not exceed 0.2%. The balance accuracy was ±0.001 g, and temperature control precision was ±2 °C.

#### 2.2.2. Determination of Composite Quality Index and Expansion Ratio of Hemp Protein Extrudate

The composite quality index (CQI) of hemp protein extrudates was assessed using a weighted scoring method, which integrates both sensory and technological parameters. The evaluation included colour, taste, odour, texture, porosity, water-holding capacity, and bulk density. Each parameter was normalized on a scale from 0 (poor quality) to 1 (excellent quality), using a reference sample with the most desirable characteristics as the benchmark.

Weighting coefficients (WiW_iWi) for each parameter were assigned based on their significance in determining the overall product quality. The final CQI was calculated using the following equation:(2)$$QL=\sum_{i=1}^{n}Wi\cdot Ki$$
where

K*i*—normalized value of the *i*-th parameter;

W*i*—weighting coefficient of the *i*-th parameter;

*n*—total number of evaluated parameters.

The weighting factors were established through expert evaluation involving five qualified specialists in food science and extrusion technology. Parameters such as texture (0.30), taste (0.25), and porosity (0.20) were assigned the highest weights due to their critical importance in consumer acceptance.

The expansion ratio (ER) of extrudates was determined as an indicator of volumetric increase upon exiting the die, reflecting the puffing and aeration behaviour of the material. The ER was calculated using the following equation:(3)$$R=\frac{D^{2}}{d^{2}}$$
where

*D*—average diameter of the extrudate after expansion (mm);

*d*—diameter of the die orifice (mm).

Measurements of the extrudate diameter were performed using a digital caliper with an accuracy of ±0.01 mm. Ten random extrudate strands were selected from each batch, and the arithmetic mean was used for analysis. The die diameter was obtained from the technical specifications of the laboratory single-screw extruder (model XYZ, Duisburg, Germany). All measurements were conducted at an ambient temperature (20–22 °C), following preliminary stabilization of the samples for 24 h in sealed polyethylene bags.

For physical-chemical evaluations, standard procedures were used in accordance with ISO 1442:1997 [18] for moisture and ISO 7971-3:2019 [19] for bulk density, where applicable.

### 2.3. Recipe and Technology of Production of Cooked-Smoked Sausages

The raw meat materials used were beef cuts (shoulder and hip), cooled to 4 ± 1 °C, as well as poultry breast fillets and fat tissue from the rear of the beef. Before processing, any external contaminants, stamps, and damaged tissue were removed from the surface of the raw materials. Deboning and trimming were carried out manually using a traditional method involving the separation of muscle tissue from bones and the removal of coarse connective tissue, cartilage, and excess fat.

The trimmed raw materials were then placed in a cold room (type SHHO-0.7) to stabilise the temperature. During preparation of the fatty tissue, any remaining connective tissue, tendons, cartilage, and films were removed. The fatty tissue was then cut into pieces weighing up to 400 g and cooled further to −1 ± 0.5 °C to ensure optimal plasticity and uniform grinding. It was then ground to a size of 3 × 3 mm using a Spiga Y2-FIA cutter (Spiga S.p.A., Milan, Italy).

The muscle raw materials (beef and chicken fillet) were pre-ground in a meat grinder with a 5–6 mm grid (model B-2) and salted. Salting was carried out using an aqueous solution of nitrite-containing salt (NaNO_2_ concentration did not exceed 2.5%), which contributed to the formation of a stable colour and ensured microbiological safety. The fat tissue was not salted and was introduced at the subsequent grinding stage.

The salted meat was then mixed with vegetable components, including hemp protein powder and sea buckthorn extract, at a ratio of 0.01% of the weight of the unsalted meat, as well as natural spices (basil, mint, nutmeg, black pepper, and sugar), and a salt mixture, in accordance with the formulation provided in Table 1. The mixture was prepared in a meat grinder (RC-40:5) for 8 min in total, including 3 min at the final stage.

The resulting minced meat was left to mature at a temperature of 2 ± 2 °C for 6–8 h to allow the formation of the product’s structure, aroma, and taste characteristics.

It was then placed in protein or artificial casings using an automatic vacuum syringe (model B6-FSB). The sausage loaves were then left to settle in a settling chamber (either a KTU-1 or a KCK-250) at a temperature of 2–4 °C and a relative humidity of 85% for 2–4 h.

After settling, the sausages underwent three consecutive stages of heat treatment. First, roasting at 80–85 °C for 65 min to form a characteristic surface and aroma. The second stage was cooking at 72–75 °C for 80 min until the internal temperature reached at least 68 °C. The final stage was hot smoking in a smoking chamber (MTU-400K) at a temperature of 30–40 °C for 45–60 min using wood smoke.

After heat treatment, the products were cooled under a wet shower for 10 min and then transferred to a refrigeration chamber (SHHO-0.7) at 10–15 °C for 30–40 min, until the internal temperature was no higher than 8 °C. The technological scheme for the production of cooked-smoked sausages with the addition of sea buckthorn extract and hemp protein is presented in Figure 1.

As can be seen in Figure 2, the control samples have a light pink hue, whereas the experimental samples have a dark pinkish-brown hue. This difference in colour is explained by the addition of sea buckthorn extract and hemp protein powder. These ingredients contain natural pigments, such as carotenoids and polyphenols, which give the product its darker colour.

Three independent batches of cooked-smoked sausages were produced for each formulation. All measurements were performed separately for each batch, and the results were expressed as mean values ± standard deviation.

### 2.4. Laboratory Analyses

Each sample was analysed within 1 h after cooking. The cooked sausages were then placed in the refrigerator (0–4 °C) without packaging and analysed after 1 and 7 days of storage.

#### 2.4.1. Determination of the pH of Meat and Meat Products

Hydrogen ion concentration was measured using the potentiometric method in a hydromodule with a 1:10 dilution [21].

#### 2.4.2. Determination of Mass Fraction of Moisture

Moisture content was determined by drying the samples to a constant weight at 103–105 °C according to the standard GOST R 51479-99 [22].

#### 2.4.3. Determination of Protein Mass Fraction

The procedure consists of the mineralization of the sample according to the Kjeldahl method, separation of ammonia by distillation into a sulphury acid solution, and subsequent titration of the tested sample (according to the standard GOST 25011-81) [23].

#### 2.4.4. Determination of Water-Holding Capacity (WHC)

The water-holding capacity (WHC) of the meat samples was determined using a modified Vartanyan method, which is based on measuring the amount of moisture released from the product under compression.

Samples of 5.00 ± 0.01 g were taken from each test batch, wrapped in two layers of filter paper (Whatman No. 1, Cytiva, Maidstone, UK), and placed between glass or metal plates. A weight of 10 kg was applied on top. Compression was carried out for 10 min at a temperature of (20 ± 2) °C [24].

After removing the load and unwrapping the sample, the filter paper was dried at 105 °C for 5 min and then weighed to determine the mass of released moisture.

The water-holding capacity (WHC) was calculated using the following formula:(4)$$\mathrm{WHC},\;\%\;=\;(1\;-\;\frac{m_{1}}{m_{0}})\times 100$$
where

*m*_0_—initial weight of the sample, g.

*m*_1_—weight of the released moisture (determined by the increase in filter paper weight), g.

A high WHC value indicates a strong ability of the tissue to retain moisture under mechanical stress, which is essential for maintaining the texture and juiciness of meat products.

Each measurement was performed in three biological replicates (different batches), with each replicate analysed in triplicate (technical replicates).

#### 2.4.5. Determination of Water-Binding Capacity (WBC)

The water-binding capacity (WBC) of the meat products was determined using a modified Vartanian method, which is based on measuring the amount of water released under applied pressure.

From each experimental batch, samples weighing 5.00 ± 0.01 g were taken, wrapped in two layers of filter paper (Whatman No. 1), and placed between two flat plates (glass or metal). A 10 kg load was applied for 10 min at a temperature of (20 ± 2) °C.

After pressing, the sample was unwrapped, and the filter paper was dried at 105 °C for 5 min. The increase in weight of the filter paper was used to determine the amount of water released from the sample.

WBC was calculated according to the following formula:(5)$$\mathrm{WBC},\;\%\;=\;(1\;-\;\frac{m_{1}}{m_{0}})\times 100$$
where

*m*_0_ is the initial weight of the sample (g);

*m*_1_ is the weight of moisture released (g).

High WBC values indicate a greater ability of the meat matrix to retain moisture under mechanical pressure, which is important for maintaining product texture and juiciness.

All measurements were performed in three biological replicates (different batches), and each was analysed in triplicate.

#### 2.4.6. Determination of Fat-Holding Capacity (FHC)

The fat-holding capacity (FHC) of the cooked-smoked sausage samples was evaluated using a gravimetric method with slight modifications based on previously established protocols.

Approximately 5.00 ± 0.01 g of each minced sample was mixed with 10.00 g of refined sunflower oil in a pre-weighed centrifuge tube. The mixture was vortexed for 60 s to ensure uniform contact and then allowed to stand at room temperature (20 ± 2 °C) for 30 min to facilitate fat absorption. Following incubation, the samples were centrifuged at 3000 rpm for 15 min. The supernatant (unabsorbed oil) was decanted and weighed.

The fat-holding capacity was calculated using the following formula:(6)$$\mathrm{FHC}(\%)\;=\;(\frac{m_{0}-m_{r}}{m_{s}})\times 100$$
where

*m*_0_ is the initial mass of oil added (g);

*m_r_* is the residual mass of oil removed after centrifugation (g);

*m_s_* is the mass of the sample (g).

The fat-holding capacity (FHC) was expressed as grams of retained fat per gram of dry matter of the sample (g/g). All measurements were performed in three biological replicates and three analytical replicates for each batch. To ensure accuracy and reproducibility, the analysis was conducted under standardized conditions: temperature, type of oil, contact time, and centrifugation parameters were kept consistent across all samples.

#### 2.4.7. Determination of Organoleptic Parameters

Experimental samples were analysed by organoleptic indicators according to the standard GOST 15115.3-77 [25].

#### 2.4.8. Determination of Fatty Acid Composition

To study the fatty acid composition, lipids were extracted from the experimental samples using hexane in a Soxhlet apparatus. The obtained extract was evaporated to dryness in a round-bottom flask using a rotary evaporator at a bath temperature of 30–40 °C. Then, 10 mL of hexane, 400 µL of 0.5 M sodium ethylate in ethanol, and 50 µL of acetic acid were added to the flask. The mixture was vigorously stirred for 2 min, left to stand for 5 min, and then filtered through a paper filter. The resulting solution was used for analysis. After derivatization to ethyl esters, the fatty acid composition was determined by gas chromatography using a Crystal 4000 gas-liquid chromatograph equipped with a flame ionization detector and the NetChrom (version 1.5, Chromatec, Russia) software [26].

#### 2.4.9. Determination of Water-Soluble Vitamins in Raw Materials and Food Products by Capillary Zone Electrophoresis Method

Capillary zone electrophoresis is a modern analytical method used to determine the mass fraction of water-soluble vitamins. It is based on the migration and separation of ionic forms of analytes under the influence of an electric field, with detection at a wavelength of 200 nm based on their electrophoretic mobility.

During sample preparation, extraction was carried out using an aqueous solution of sodium tetraborate in the presence of sulphite ions. The resulting extract was centrifuged at 5000–6000 rpm for 5 min and, if necessary, filtered through a membrane filter.

This method is primarily used to determine the mass fraction of free vitamin forms in premixes, vitamin additives, concentrates, and mixtures. In this study, the appropriate sample mass of the test suspensions was selected to ensure accurate quantification.

#### 2.4.10. Determination of Amino Acid Composition

The mass fraction of amino acids in the analysed products was determined on the «Kapel 105 M» (Lumex, Russia) capillary electrophoresis system. The method is based on the decomposition of samples by acid or alkaline hydrolysis of amino acids into free forms, in which obtaining phenylisothiocarbamyl derivatives are obtained and are further separated and quantified by capillary electrophoresis. Detection is carried out in the UV region of the spectrum at a wavelength of 254 nm. Detection and further data processing were carried out on the software «Elforan» (version 3.1, Lumex, Russia).

Mass fraction of amino acid (X, %) is calculated according to the following formula:(7)$$X=\frac{100*V_{hydr}*V_{final}*C_{meas}}{1000*m*V_{aliq}}$$

*X*—mass fraction of amino acid in the sample, %;

C_meas_—measured value of amino acid mass concentration in the prepared solution, mg/dm^3^;

*m*—sample weight, mg;

V_hydr_—total volume of hydrolysate, cm^3^;

V_final_—volume of the analysed solution, cm^3^;

V_aliq_—volume of aliquot portion of hydrolysate, cm^3^;

100—multiplier for expressing results as a percentage;

1000—volume unit dimensionality reconciliation factor.

#### 2.4.11. Assessment of Amino Acid Composition and Calculation of Protein Biological Value

The amino acid composition of cooked-smoked sausages was determined by ion-exchange chromatography with post-column ninhydrin derivatization using an automatic amino acid analyser AAA-400 (INGOS, Prague, Czech Republic). Hydrolysis was carried out in 6 N HCl at 110 °C for 24 h in sealed ampoules under nitrogen. For the determination of sulphur-containing amino acids (methionine and cystine), preliminary oxidation with formic acid and hydrogen peroxide was applied. The results are expressed in milligrams per 100 g of product (mg/100 g).

The biological value of the protein was assessed in silico based on the amino acid composition by comparative analysis with the reference protein proposed by FAO/WHO (2007) [27]. The calculations included the following indicators:

Amino Acid Score (*AAS*, %):(8)$$AASi=\left(\frac{AAsample,i}{AAreference,i}\right)\times 100$$

Chemical Score (CS): the lowest value of the Amino Acid Score (*AAS*), indicating the limiting amino acid;

Essential Amino Acid Index (*EAAI*, %): calculated according to the Oser formula,(9)$$EAAI=(\prod_{i-1}^{n}\frac{AAsample,i}{AAreference,i})\times 100\;$$
where *n* is the number of essential amino acids.

The calculations were performed using spreadsheet software based on the amino acid composition data. The amino acid profile for adults proposed by FAO/WHO (2007) [27] was used as the reference standard.

#### 2.4.12. Determination of Microbiological Parameters

Microbiological analysis was performed in accordance with GOST 26668-85 [28], GOST 26669-85 [29], GOST 26972-85 [30], and GOST 26670-85 [31]. To identify *Escherichia coli*, *Staphylococcus aureus*, and sulphite-reducing clostridia, selective culture media manufactured by Sigma-Aldrich (Burlington, MA, USA) were used in accordance with the manufacturer’s instructions.

#### 2.4.13. Statistical Analysis

Statistical analysis was performed in order to evaluate the effect of the hemp protein and sea buckthorn extract on the quality and shelf life of cooked-smoked sausages by one-way ANOVA and Tukey HSD test as posthoc tests, using Excel and SPSS (Statistics version 27) for Windows V27.0.1.0 software (SPSS, Inc., Chicago, IL, USA, 2020). All measurements were carried out in triplicate (*n* = 3), and the results are presented as mean ± standard deviation. Results were considered statistically significant at a *p*-level equal to or less than 0.05 (*p* < 0.05). Analyses of the graphical dependencies of the evaluation criteria on the studied factors were performed using SPSS for Windows V27.0.1.0 software (SPSS, Inc., Chicago, IL, USA, 2020).

## 3. Results

### 3.1. Composite Quality Index and Expansion Ratio of Hemp Protein Extrudate

Based on the graphical dependencies of the evaluation criteria on the investigated factors (Figure 3), the following conclusions can be made. The comprehensive quality index and the expansion coefficient of the extrudate exhibit similar trends. It was found that both parameters initially increase with increasing screw speed and initial moisture content of the product, reach peak values, and then gradually decline. This behaviour is attributed to the fact that, at low screw speeds, the melt remains in the pre-matrix zone for a longer duration under high temperature and pressure conditions. This can lead to partial thermal degradation of starch granules and sintering of the melt, which reduces expansion due to loss of elastic recovery forces and hinders structure formation during explosive moisture evaporation [32]. Conversely, excessively high screw speeds can result in the accumulation of excess thermal energy, causing similar negative effects. Furthermore, adjusting the moisture content of the raw material—either increasing or decreasing it—can influence product quality. The specific productivity is determined by the extruder’s throughput and the total energy consumption under steady-state conditions. An increase in the initial moisture content decreases the viscosity of the melt, thereby enhancing productivity.

### 3.2. Nutritional and Biological Values of Cooked-Smoked Sausages

The results of the evaluation of quality indicators of a new type of cooked-smoked sausage that has the addition of sea buckthorn and hemp protein extracts are presented in the table.

The results of the analysis in Table 2 show that the finished products that have the addition of sea buckthorn and hemp protein extracts meet the requirements for cooked-smoked sausages by organoleptic and physicochemical parameters.

The addition of extracts to the formulation of cooked-smoked sausage has a significant increase in the mass fraction of protein in the finished product. Due to the energy overload of the human diet, the reduction in fat content is a positive factor.

Further the content of biologically active substances in the finished product was investigated (Table 3).

By the number of vitamins, the new formulation slightly exceeds the content of thiamine, riboflavin, and pyridoxine. At the same time, the introduction of a vegetable additive enriches the developed sausages with carotenoids, tocopherol, and flavonoids, which, of course, increases their biological value and transfers them to the range of products of special and functional foods [33].

The peculiarity of the physiological role of β-carotene and tocopherol consists in their antioxidant interference in the processes of lipid peroxidation in the organism, mainly in membrane structures (embedded by the side chain in the membrane structure), which contributes to the preservation of the quality of finished products [34].

Data from Table 3 show that the control sample provides about 20% of the daily requirement of the human body in pyridoxine and β-carotene, more than 10% of the daily requirement in thiamine and tocopherol, and about 8% in riboflavin. By adding 0.01% of the extracts to the recipe of cooked-smoked sausage, the tocopherol content increased 2.5 times.

The calculated degree of satisfaction of the average daily physiological requirement of the human body in polyunsaturated fatty acids when consuming 100 g of samples of that cooked-smoked meat product was more than 10%.

The presented research results allow us to attribute cooked-smoked sausage to the products of special or functional purpose, because the consumption of 100 g of these products provides more than 10% of the average daily physiological requirement of the human body in polyunsaturated fatty acids, riboflavin, thiamine, pyridoxine, and tocopherol. The developed recipe of cooked-smoked sausage allows it to expand into the range of cooked-smoked sausages of high nutritional and biological value.

### 3.3. Amino Acid Composition of Cooked-Smoked Sausages

The amino acid composition of cooked-smoked sausages, including experimental and control samples, was studied. The obtained results are presented in Table 4.

Besides the amino acid composition of the boiled-smoked sausages in the control and experimental samples (mg/100 g of product), the biological values of the proteins in the investigated products were also evaluated. The results of calculations are presented in Table 5 and Table 6.

The Amino acid contents were recalculated per 100 g of the protein (mg/100 g protein) in the control and experimental samples.

The proteins’ biological values, based on amino acid composition, were evaluated (calculated per 100 g protein).

A direct correlation was observed between the increase in amino acid content and the biological value of protein in the final product, depending on the proportion of plant-based ingredients incorporated into the formulation. In the experimental samples of cooked-smoked sausages, threonine was identified as the limiting amino acid. Replacing 0.01% of meat with plant-based components resulted in a 2.48% increase in the biological value of protein and a 17.9% improvement in the Amino Acid Score (Cj) of threonine. Compared to the control, the prototype showed enhanced protein quality. Except for threonine, the Cj values of all other essential amino acids exceeded 100% in the experimental formulation.

### 3.4. Organoleptic Characteristics of Boiled and Smoked Sausages with the Addition of Sea Buckthorn Extract and Hemp Protein

In accordance with the developed technology, an experimental batch of boiled and smoked sausages was made with the introduction of sea buckthorn extracts and hemp protein, and its organoleptic parameters were studied. The control sample was boiled and smoked sausage without extracts (Figure 4).

The tasting evaluation of boiled and smoked sausages with the introduction of sea buckthorn seed powder showed high organoleptic characteristics of the finished product.

The results in Figure 4 of the organoleptic evaluation of boiled and smoked sausages showed that the prototypes with a 0.01% administration of sea buckthorn extracts and hemp protein had an attractive original appearance on the section with granular inclusions. The colour on the cross-section of the samples was uniformly dark pinkish-brown, without grey areas and consistent throughout the entire loaf mass. The smell peculiar to boiled and smoked sausages is pleasant. The taste is moderately salty, characteristic, and without extraneous flavours. The consistency of the prototypes is quite elastic and moderately tender. Based on the conducted studies, high organoleptic characteristics of finished sausage products that have sea buckthorn extracts and hemp protein, in an amount of 0.01%, have been established.

### 3.5. Microbiological Parameters of the Finished Product During Storage

Studies were conducted to assess the microbiological safety indicators of cooked and smoked sausages during storage. The presences of *Escherichia coli* (*E. coli*) and *Staphylococcus aureus* (*S. aureus*) were determined, as was the number of mesophilic aerobic and facultative anaerobic microorganisms and sulphite-reducing clostridia. The finished products were stored in vacuum packaging at a temperature of 4 ± 1 °C. During the first 36 days of storage, all samples complied with the requirements of TR CU 034/2013 [35]; ‘On the safety of meat and meat products’, the specified pathogens were not detected, and total microbial contamination levels were within acceptable limits.

After day 36, there was a tendency for the total number of mesophilic aerobic and facultative anaerobic microorganisms to increase, indicating a decrease in the product’s microbiological stability. The samples were stored for up to 40 days as part of an experimental observation to determine the maximum storage period. However, this is not recommended from an industrial point of view. The results of these studies are presented in Table 7 and Table 8.

Thus, based on the data obtained, it can be concluded that the shelf life of cooked-smoked sausages prepared with the addition of hemp proteins and sea buckthorn extracts is up to 36 days at a storage temperature of 4 ± 1 °C and in airtight packaging. Fungal contamination was not assessed in this study, which is a limitation of the current microbiological analysis. Industrial approbation and its introduction into production was carried out at the enterprise Pervomayskiye Delicatessen LLP.

## 4. Discussion

The study analysed the relationship between screw speed, initial moisture content of the extruded raw material, and extrudate quality indicators. The results confirmed that, with the increase in these parameters, the complex quality index and the coefficient of extrudate expansion first increased up to a certain limit, after which there was a decrease [36]. This is explained by partial thermal decomposition of the microstructure of starch grains and sintering of a part of the melt, which prevents the expansion of the extrudate due to limited elastic recovery of the structure during moisture evaporation [37]. Similar patterns have been found in the works of other researchers. For example, Shirazi et al. [38] demonstrated that an increase in raw material moisture content during extrusion of barley and carrot pomace leads to a decrease in the coefficient of extrudate expansion. This is in agreement with our results, where it is also shown that increased raw material moisture content leads to a decrease in melt viscosity and a decrease in product hardness. Similarly, Hashemi et al. [39] observed that high moisture content of extruded materials reduces the density and hardness of the final product, which is consistent with the patterns we have identified in extruding starch-containing materials. In addition, Singh et al. [40] investigated the effect of moisture content, screw speed, and drum temperature on the properties of potato-starch-based extrudates. They concluded that an increase in moisture content decreases energy input and water solubility index but increases density and the water absorption index. These data confirmed our results, which also observed a decrease in specific mechanical energy and the water solubility index with increasing moisture content. The study of the effect of sea buckthorn extracts and hemp protein on the nutritional and microbiological characteristics of cooked-smoked sausages showed that the addition of these ingredients significantly increases the content of protein and tocopherols in the finished product, which contributes to an increase in its nutritional value and antioxidant activity. This is supported by the research of Tokysheva et al. [41], who showed that the addition of protein hydrolysate improves the protein composition of sausages. Similar results were obtained by Ghribi et al. [42] who observed that the addition of chickpea protein increases the protein content of processed meat products, which is also in agreement with the study of authors Bukyei, E et al. [43]. When cooked-smoked sausages were enriched with four different amounts of sea buckthorn peel, 0.2%, 0.3%, 0.4% and 0.5%, compared to control samples, the amounts of flavonoids, calcium, and magnesium increased significantly. Microbiological studies confirmed that the use of sea buckthorn and hemp protein extracts extended the shelf life of the products due to natural antioxidants. These results are in agreement with the study of Alirezalu et al. [44], where green tea and nettle extracts contributed to an increase in the shelf life of sausages. Thus, the use of natural additives in meat products not only improves their nutritional value but also enhances safety by extending shelf life [45]. In conclusion, the regularities of interaction between technological parameters of extrusion and raw material composition and their influence on the qualitative characteristics of products revealed in this study are confirmed by the studies of other authors. These data can be used to optimise extrusion conditions in order to achieve the best quality of finished products. The results obtained are consistent with the studies by the authors [46], who showed that optimising the technological parameters of plant raw materials, such as sprouted buckwheat, contributes to increasing the nutritional and functional value of products. Similarly, the addition of hemp proteins and sea buckthorn extracts in this study led to an increase in the protein and antioxidant content of cooked-smoked sausages, confirming the potential of using plant components to develop high-quality functional meat products.

The impact of hemp-based ingredients on meat products was also reported by Wang et al. [47]. The addition of 5% hemp protein and other hemp ingredients to pork loaves increased fibre and mineral content but negatively affected sensory qualities, especially with hemp protein and flour. Only loaves with de-hulled hemp seeds showed acceptable taste. These findings support our observations and highlight the importance of balancing nutritional benefits with sensory acceptability [47].

Bozhko et al. evaluated the addition of hemp cake (3–7.4%) in pork meatballs during refrigerated storage. They observed a significant reduction in protein and lipid oxidation (up to 11% and 62%, respectively) and acceptable sensory scores at lower inclusion levels (~0.9%). However, higher concentrations led to deterioration in the taste, colour, and juiciness of the products [48].

Bozhko et al. also investigated meat-containing bread enriched with 8–12% hemp flour. The enriched samples exhibited enhanced water- and fat-holding capacities (by 10–22% and 17–27%, respectively), increased protein and mineral content, and maintained acceptable sensory properties at an inclusion level of approximately 10% [49]. The addition of 3% juniper berries led to a two-fold increase in bread’s antioxidant activity (15.5 mg/100 g), contributing to extended shelf life and freshness retention [50].

## 5. Conclusions

The study demonstrated that changes in screw speed and initial moisture content had a statistically significant effect on the quality of extruded protein ingredients. An increase in these parameters led to a rise in the expansion ratio and the overall comprehensive quality index up to a certain point, after which the indicators declined. Higher initial moisture content reduced melt viscosity, thereby enhancing the efficiency of the extrusion process.

The addition of 0.01% hemp proteins and sea buckthorn extracts to the formulation of cooked-smoked sausages contributed to an improved nutritional value. An increase in total protein content, a 2.5-fold rise in tocopherol levels, a 2.48% increase in biological protein value, and a 17.9% increase in the Amino Acid Score for threonine were observed. These results indicate better biological performance of the experimental samples compared to the control group. The experimental sausages exhibited enhanced organoleptic characteristics, including an original structure, more intense colour, and attractive appearance. The presence of tocopherols, β-carotene, and flavonoids—natural antioxidants—suggests potential antioxidant activity of the product; however, further physicochemical investigations are required to confirm this. Microbiological analysis during 40 days of storage revealed no presence of pathogenic microorganisms. Total microbial counts remained within acceptable limits for up to 36 days, supporting the recommended shelf life under chilled conditions (0–4 °C). Thus, the developed cooked-smoked sausage formulation with 0.01% sea buckthorn extract and hemp protein enables the expansion of the range of functional meat products with improved nutritional and biological properties.

## Figures and Tables

**Figure 1 foods-14-02730-f001:**
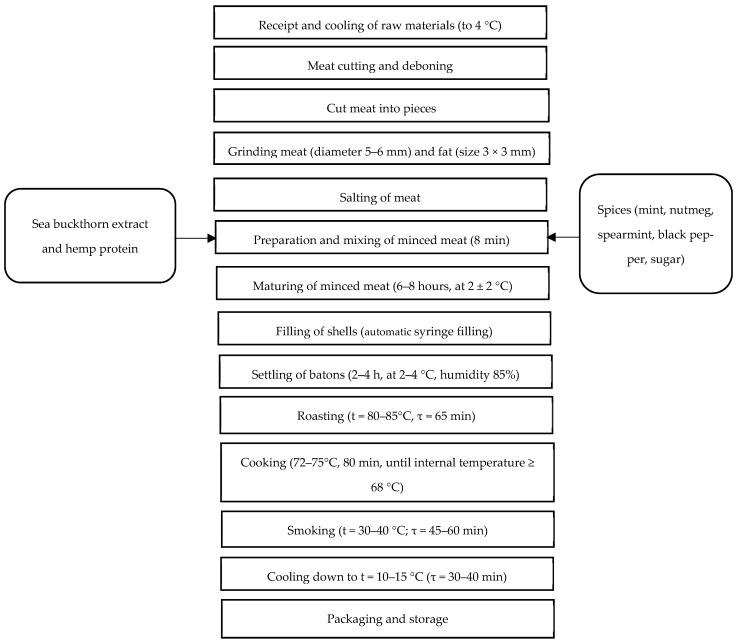
Technological scheme of cooking cooked-smoked sausages prepared with the addition of sea buckthorn extract and hemp protein [20].

**Figure 2 foods-14-02730-f002:**
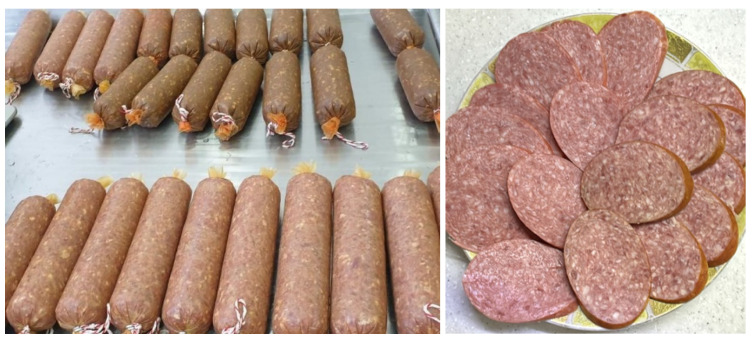
Boiled-smoked sausage prepared with the addition of sea buckthorn extract and hemp protein powder.

**Figure 3 foods-14-02730-f003:**
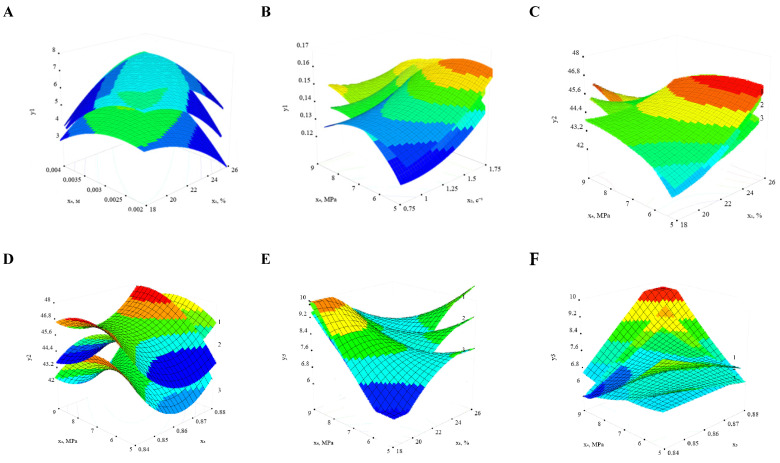
Expansion coefficient of hemp protein extrudate. (**A**) Dependence of specific energy consumption on initial product moisture content and pressure in the pre-matrix zone at different screw speeds x_2_, s^−1^: 1—0.9; 2—1.15; 3—1.4 and x_3_ = 0.86. (**B**) Dependence of specific energy consumption on the design parameter and pressure in the pre-matrix zone at different values of initial humidity of products x_1_, %: 1—18; 2—22; 3—26 and x_2_ = 1.15 s^−1^. (**C**) Dependence of specific productivity on initial product moisture content and pressure in the pre-matrix zone at different screw speeds x_2_, s^−1^: 1—0.9; 2—1.15; 3—1.4 and x_3_ = 0.86. (**D**) Dependence of specific productivity on the design parameter and screw rotation speed at different pressures in the pre-matrix zone x_4_, MPa: 1—5; 2—7; 3—9 and x_1_ = 22%. (**E**) Dependence of the comprehensive quality indicator on initial product moisture content and screw speed at different pressures in the pre-matrix zone x_4_, MPa: 1—5; 2—7; 3—9 and x_3_ = 0.86. (**F**) Dependence of the comprehensive quality indicator on pressure in the pre-matrix zone and design parameter of the initial moisture content of the products, %: 1—18; 2—22; 3—26 and x_2_ = 1.15 s^−1^.

**Figure 4 foods-14-02730-f004:**
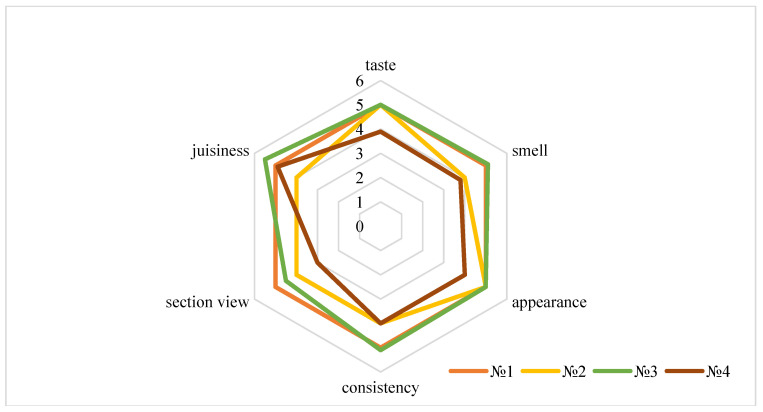
Organoleptic characteristics of boiled and smoked sausages with the addition of sea buckthorn extracts (№ 1—control sample; № 2—with the addition of a 0.005% extract of sea buckthorn and hemp protein; № 3—with the addition of a 0.01% extract of sea buckthorn and hemp protein; № 4—with the addition of a 0.015% extract of sea buckthorn and hemp protein). Note the following: Organoleptic scores are presented as mean values. Due to graphical constraints, standard deviation values (SD) are not shown but are taken into account in the analysis. The evaluations were conducted by five trained panellists, and statistical significance was assessed using appropriate methods.

**Table 1 foods-14-02730-t001:** Recipe of cooked and smoked sausages [20].

Ingredients	Quantity
100 kg meat raw material, %	
Fillet beef	60
Poultry fillet	20
Beef fat	20
non-meat raw materials	
Table salt, %	2.5
Sodium nitrite, g/100 kg	0.055
Hemp protein powder, g/100 kg	10
Sea buckthorn extract, g/100 kg	10
Sugar, g/100 kg	100
Black pepper, g/100 kg	85
Nutmeg, g/100 kg	55
Basil, g/100 kg	3
Mint, g/100 kg	3

**Table 2 foods-14-02730-t002:** Quality indicators of cooked-smoked sausage.

Indicators	With Addition of Sea Buckthorn Extract and Hemp Protein	Control
Mass fraction, %		
Moisture	60.48 ± 0.05 ^a^	63.66 ± 0.05 ^b^
Protein	25.23 ± 0.05 ^b^	22.75 ± 0.08 ^a^
Fat-soluble antioxidants, mg/g	0.08 ± 0.0004 ^a^	0.11 ± 0.0010 ^b^
Water-soluble antioxidants, mg/g	0.49 ± 0.0034 ^b^	0.36 ± 0.0038 ^a^
Water-holding capacity, %	60.46 ± 0.02 ^a^	63.16 ± 0.02 ^b^
Water-binding capacity, %	59.60 ± 0.02 ^a^	56.84 ± 0.05 ^b^
Fat-holding capacity, %	54.18 ± 1.02 ^a^	57.21 ± 0.85 ^b^

Note: Values are expressed as mean ± standard deviation (*n* = 3). Means within the same row followed by different superscript letters (a, b) are significantly different (*p* < 0.05).

**Table 3 foods-14-02730-t003:** Content of biologically active substances in finished products.

Indicators	Daily Requirement	With Addition of Sea Buckthorn Extract and Hemp Protein	Control
Vitamin composition, mg/100 g product			
Polyunsaturated fatty acids, %	11 mg/day	1.35 ± 0.028 ^b^	1.22 ± 0.023 ^a^
Pyridoxine (B6)	1.8–2.0 mg/day	0.39 ± 0.02	0.35 ± 0.03
Riboflavin (B2)	1.8 mg/day	0.187 ± 0.004	0.180 ± 0.005
Thiamine (B1)	1.5 mg/day	0.283 ± 0.003 ^b^	0.250 ± 0.004 ^a^
Tocopherol	7.8 mg/equivalent day	0.705 ± 0.0265 ^b^	0.301 ± 0.0144 ^a^
Flavonoids, mg/100 g	–	0.13 ± 0.08	–
Carotenoids, mg/100 g	–	0.19 ± 0.02	–

Note: Values are expressed as mean ± standard deviation (*n* = 3). Means within the same row followed by different superscript letters (a, b) are significantly different (*p* < 0.05).

**Table 4 foods-14-02730-t004:** Amino acid composition of cooked-smoked sausages.

Time	Component	Control Sample—Concentration (mg/100 g)	Control Sample—Mass Fraction of Amino Acids (%)	Experimental Sample—Concentrations (mg/100 g)	Experimental Sample—Amino Acid Mass Fraction (%)	*p*-Value
6.188		0.00	0.00	0.00	0.00	
6.285	arginine	110.0 ± 0.088	2.422 ± 0.096	93.0 ± 0.799	2.000 ± 0.800	*p* < 0.05
8.288	lysine	120.0 ± 0.799	2.642 ± 0.898	100.0 ± 0.453	2.151 ± 0.731	*p* < 0.05
8.540	tyrosine	43.0 ± 0.212	0.947 ± 0.284	39.0 ± 0.197	0.839 ± 0.252	*p* < 0.05
8.653	phenylalanine	56.0 ± 0.298	1.233 ± 0.370	53.0 ± 0.278	1.140 ± 0.342	*p* < 0.05
8.873	histidine	43.0 ± 0.398	0.947 ± 0.473	37.0 ± 0.212	0.796 ± 0.398	*p* < 0.05
9.182	leucine + isoleucine	93.0 ± 0.232	2.048 ± 0.532	86.0 ± 0.361	1.849 ± 0.481	*p* < 0.05
9.340	methionine	45.0 ± 0.235	0.991 ± 0.337	37.0 ± 0.193	0.796 ± 0.271	*p* < 0.05
9.448	valine	97.0 ± 0.623	2.136 ± 0.854	96.0 ± 0.366	2.065 ± 0.826	*p* > 0.05
9.598	proline	70.0 ± 0.287	1.541 ± 0.308	72.0 ± 0.302	1.548 ± 0.403	*p* > 0.05
9.703	threonine	67.0 ± 0.389	0.417 ± 0.401	55.0 ± 0.334	1.183 ± 0.473	*p* < 0.05
10.015	serine	56.0 ± 0.456	1.475 ± 0.590	44.0 ± 0.265	0.946 ± 0.246	*p* < 0.05
10.145	alanine	100.0 ± 0.510	2.202 ± 0.572	97.0 ± 0.456	2.086 ± 0.542	*p* > 0.05
10.655	glycine	84.0 ± 0.567	1.849 ± 0.629	90.0 ± 0.612	1.935 ± 0.658	*p* > 0.05

Note: Values are presented as mean ± standard deviation; *n* = 3.

**Table 5 foods-14-02730-t005:** Results of the calculation of the biological value of proteins in the control samples (boiled-smoked sausages).

Amino Acids	Indicators							
	Content, mg/100 g	Aj, g/100 g	Acj, g/100 g	Cj, %	Δ Amino Acid Availability Deviation, %	Coefficient of Rationality of Amino Acid Score, %	Biological Value, %	Aj
lysine	865 ± 24	5.15	5.5	93.6	33.3	22.34	77.6	0.64
tyrosine + phenylalanine	1021 ± 31	6.08	6.0	101.3	41.0			0.60
leucine + isoleucine	1616 ± 45	9.62	11.0	87.5	27.2			0.69
methionine	458 ± 18	2.73	3.5	78.0	17.7			0.77
valine	746 ± 21	4.44	5.0	88.8	28.5			0.68
threonine	405 ± 17	2.41	4.0	60.3	-			1
tryptophan	116 ± 9	0.69	1.0	69.0	8.7			0.87

Note: Values are presented as mean ± standard deviation; *n* = 3.

**Table 6 foods-14-02730-t006:** The results of calculating the biological value of proteins in a sample of boiled and smoked sausage.

Amino Acids	Indicators							
	Content, mg/100 g	Aj, g/100 g	Acj, g/100 g	Cj, %	Δ Amino Acid Availability Deviation, %	Coefficient of Rationality of Amino Acid Score, %	Biological Value, %	Aj
lysine	1152 ± 28	6.2	5.51	112.7	28.9	19.9	80.1	0.75
tyrosine + phenylalanine	1232 ± 37	6.62	6.1	110.3	26.5			0.77
leucine + isoleucine	2195 ± 48	11.79	11.1	107.2	23.4			0.79
methionine	685 ± 21	3.65	3.51	104.2	20.3			0.82
valine	1009 ± 28	5.42	5.1	108.3	24.5			0.79
threonine	626 ± 20	3.36	4.1	83.9	-			1.1
tryptophan	190 ± 12	1.1	1.1	100.1	16.3			0.85

Note: Values are presented as mean ± standard deviation; *n* = 3.

**Table 7 foods-14-02730-t007:** Changes in the microflora during the storage of the finished products.

Name of Indicators, Units of Measurement, 1 g, log CFU/g	Actual Results				
	Regulatory Limit	1	2	3	4
		1 Days	15 Days	36 Days	40 Days
The number of mesophilic aerobic and facultative anaerobic microorganisms, colony-forming unit	≤3.30	2.00 ± 0.05	2.60 ± 0.06	2.78 ± 0.07	3.15 ± 0.04
*E. coli*	ND	ND	ND	ND	ND
*S. aureus*	ND	ND	ND	ND	ND
Sulphite-reducing clostridium	ND	ND	ND	ND	ND

Note: Values are presented as mean ± standard deviation; *n* = 3.

**Table 8 foods-14-02730-t008:** Changes in the microflora during the storage of the control sample.

Name of Indicators, Units of Measurement, 1 g, log CFU/g	Actual Results				
	Regulatory Limit	1	2	3	4
		1 Days	15 Days	36 Days	40 Days
The number of mesophilic aerobic and facultative anaerobic microorganisms, colony-forming unit	≤3.30	2.48 ± 0.06	2.85 ± 0.05	2.95 ± 0.06	3.48 ± 0.05
*E. coli*	ND	ND	ND	ND	ND
*S. aureus*	ND	ND	ND	ND	ND
Sulphite-reducing clostridium	ND	ND	ND	ND	ND

Note: Values are presented as mean ± standard deviation; *n* = 3.

## Data Availability

The original contributions presented in this study are included in the article. Further inquiries can be directed to the corresponding author.

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
