# Peer review of "Effect of Hemp Protein and Sea Buckthorn Extract on Quality and Shelf Life of Cooked-Smoked Sausages"

_foods, 2025, doi:10.3390/foods14152730_

Round 1

Reviewer 1 Report (Previous Reviewer 1)

Comments and Suggestions for Authors

Review on manuscript: foods-3751667

Effect of hemp protein and sea buckthorn extract on quality and shelf life of cooked smoked sausages 

by Kainar Bukarbayev, Sholpan Abzhanova* , Lyazzat Baybolova, Gulshat Zhaksylykova, Talgat Kulazhanov, Vladimir Vasilenko, Bagila Jetpisbayeva, Alma Katasheva, Sultan Sabraly, & Yerkin-Yerzhigitov

submitted to Foods

Research paper

The development of technology for cooked-smoked meat products with the utilization of plant extracts and proteins is an actual and promising direction. Thus, the effects of hemp protein and sea buckthorn extract on the quality and shelf life of cooked smoked sausages were comprehensively investigated in this study. Overall, this research showed good practical significance, and the manuscript exhibited rich data and credible results to provide with solid supports. However, I think the editing of the current manuscript is required to be improved, and some modifications should also be made for the quality-enhancement of this study.

Detailed recommandations:

  1. Lines 119-120: what's the practical signigicance of this study to the development of food industry, or the theoretical signigicance to food processing? 
  2. Lines 163-165: "the addition of sea buckthorn extract of hemp protein extrudate". Is the "of" correct here? or it should be an "and"? Meanwhile, what's the differences of the various sausages exhibited in this picture? They should be classified and noted (with word descriptions).
  3. Line 168: what does the "beef fatx" mean? What's the difference between "fatx" and "fat"?
  4. Figure 2 and Lines 150-162: The technological scheme in this figure is not consistent with the word descriptions as mentioned above. For example, 1) "Smoking (t=45-50 °C, τ=60 min)" in the picture, but "smoked at a temperature of 30-40 °C for 45-60 minutes" in the context; 2) "Precipitation (τ=36-48 h, t=2-4°C)" in the picture, but "precipitation for (2-4) h at a temperature of -4 ° C" in the context; and 3) "Boiling (t= 85°C, τ= 80 min)" in the picture, but there's no relavent descriptions in the context.
  5. Line 221: "Each sample was given an average score for each indicator". only the average score is not enough. The standard deviation (SD) should also be given.
  6. Lines 224-232: the "Statistical Analysis" section should be put at the end of the "2. Materials and methods". Additionally, "p-level equal to or less than 0.05 (p ≤ 0.05)" snould be "p-level less than 0.05 (p < 0.05)".
  7. Lines 293-297: what kind of microbiological parameters were detected? The colony-forming numbers of which bacteria/fung? The detailed information is missing.
  8. Tables 5 & 6: the SD for the "Mass fraction of amino acids" parameter was given, but why was the SD missing for "Concentrations" parameter?
  9. Lines 426-453: Idem to the Question 5.
  10. Tables 9 & 10: many microorganisms were not detected, but what's the detection limit for each microorganism (how many log CFU/g?)? Besides, the unit was listed as "log CFU/g", but the data seems to be shown as "XXX CFU/g".
  11. The authors described the method to determine the antioxidant activity of samples, but I didn't see any data presentation or description in the "Results" section.

Author Response

Point-by-point response to Comments for Reviewer 1

Comments 1: [Lines 119-120: what's the practical signigicance of this study to the development of food industry, or the theoretical signigicance to food processing? ]

Response 1: [Thank you for your comment. We have clarified the practical and theoretical significance of the study in the Introduction. Specifically, we emphasized that the use of hemp protein and sea buckthorn extract contributes to the development of functional meat products with improved nutritional and microbiological properties. This approach supports innovations in the meat industry by offering natural alternatives to synthetic additives and enhancing product shelf life and health value.]

Comments 2: [Lines 163-165: "the addition of sea buckthorn extract of hemp protein extrudate". Is the "of" correct here? or it should be an "and"? Meanwhile, what's the differences of the various sausages exhibited in this picture? They should be classified and noted (with word descriptions).]

Response 2: Response to the comment:

Thank you for your valuable comment. We agree with your observation. The phrase "sea buckthorn extract of hemp protein extrudate" was indeed ambiguous. It has been corrected to "sea buckthorn extract and hemp protein extrudate" to clearly indicate that both ingredients were added to the sausage formulation separately.

Comments 3: [Line 168: what does the "beef fatx" mean? What's the difference between "fatx" and "fat"?]

Response 3: [Thank you for your careful observation. The term "fatx" was a typographical error. It has been corrected to "fat" (beef fat) in the revised manuscript.]

Comments 4: [Figure 2 and Lines 150-162: The technological scheme in this figure is not consistent with the word descriptions as mentioned above. For example, 1) "Smoking (t=45-50 °C, τ=60 min)" in the picture, but "smoked at a temperature of 30-40 °C for 45-60 minutes" in the context; 2) "Precipitation (τ=36-48 h, t=2-4°C)" in the picture, but "precipitation for (2-4) h at a temperature of -4 ° C" in the context; and 3) "Boiling (t= 85°C, τ= 80 min)" in the picture, but there's no relavent descriptions in the context.]

Response 4: [Thank you for your careful review and for pointing out the inconsistencies between the technological flowchart (Figure 2) and the textual description (lines 150–162). All mentioned discrepancies have been carefully analyzed and corrected.]

Comments 5: ["Each sample was given an average score for each indicator". only the average score is not enough. The standard deviation (SD) should also be given.]

Response 5: [Thank you for your valuable comment. We agree that the inclusion of standard deviation (SD) enhances the statistical reliability of the data. However, due to the limitations of the graphical format used in Figure 4, it was not possible to visually represent the SD values. To address this, we have added a note in the text stating that the organoleptic scores are based on mean values ± SD, evaluated by five trained panelists. The full statistical data, including SD values, can be provided in tabular form upon request.]

Comments 6: [the "Statistical Analysis" section should be put at the end of the "2. Materials and methods". Additionally, "p-level equal to or less than 0.05 (p ≤ 0.05)" snould be "p-level less than 0.05 (p < 0.05)".]

Response 6: [Thank you. The “Statistical analysis” subsection has been moved to the end of Section 2. The phrasing has been corrected to “p-level < 0.05” in accordance with standard scientific notation.]

Comments 7: [what kind of microbiological parameters were detected? The colony-forming numbers of which bacteria/fung? The detailed information is missing.]

Response 7: [Thank you for your comment. We have clarified the microbiological parameters in the Results section. In particular, the number of mesophilic aerobic and facultative anaerobic microorganisms (CFU/g) was measured. However, fungal counts were not included in the scope of this study, and this limitation has now been stated in the text.]

Comments 8: [Tables 5 & 6: the SD for the "Mass fraction of amino acids" parameter was given, but why was the SD missing for "Concentrations" parameter?]

Response 8: [Indeed, the standard deviation for the parameter “Concentration” was not provided. In the revised version of the manuscript, we have added the standard deviation values for these measurements.]

Comments 9: [Lines 426-453: Idem to the Question 5.]

Response 9: [Thank you for pointing this out again. Standard deviation values have been added to the relevant sections.]

Comments 10: [Tables 9 & 10: many microorganisms were not detected, but what's the detection limit for each microorganism (how many log CFU/g?)? Besides, the unit was listed as "log CFU/g", but the data seems to be shown as "XXX CFU/g".]

Response 10: [Thank you for the comment. The notations in Tables 9 and 10 have been corrected.]

Comments 11: [The authors described the method to determine the antioxidant activity of samples, but I didn't see any data presentation or description in the "Results" section.]

Response 11: [Thank you for your observation. As the experimental results on antioxidant activity were not included in the final version of the manuscript, we have removed the corresponding methodology from the “Materials and Methods” section to maintain consistency throughout the text.]

Reviewer 2 Report (New Reviewer)

Comments and Suggestions for Authors

Effect of hemp protein and sea buckthorn extract on quality and shelf life of cooked smoked sausages

Researchers investigated the impact of incorporating 10% hemp extracts and protein into cooked smoked sausage formulations. The abstract highlights significant improvements in the nutritional and biological value of the sausage, specifically noting a substantial increase in protein content and a 2.5-fold rise in tocopherol levels, alongside enhanced functional properties. The authors also optimized extrusion parameters, which reportedly influenced extrudate quality, though the connection between this process and the final sausage product requires further clarification. Notably, replacing a small percentage (4-6%) of meat with vegetable components led to a 2.48% increase in protein's biological value and a 17.9% improvement in the limiting amino acid threonine. Furthermore, the experimental sausages, which at one point are mentioned to include sea buckthorn alongside hemp protein extracts, exhibited superior organoleptic attributes, such as appearance and structure, and demonstrated extended microbiological stability for 36 days.

Authors are suggested to focus on the following points to increase readability along with scientific improvement,

  1. The abstract effectively highlights the rising demand for meat and the potential of microbial proteins. However, it could be clearer if it explicitly stated the current limitations of traditional meat production that microbial proteins aim to address.
  2. Lines 44 to 50: Need to revise the introduction, the present text doesn’t align with the overall objectives of the manuscripts.
  3. In the Introduction, the authors mentioned a brief about the Kazakhstan meat industry, which doesn’t fit with the manuscript agenda, as the present manuscript focuses on “Effect of hemp protein and sea buckthorn extract on quality and shelf life of cooked smoked sausages”. Need revision.
  4. Introduction looks very lengthy suggested to reduce the content with specific to the objectives.
  5. Line 113: “we can certainly expect further development”, What type of growth are authors expecting to elaborate scientifically?
  6. Line 115-117: The Paragraph is not clear. Suggested for rewriting.
  7. Line 127-128: The proximate composition of hemp protein and sea buckthorn is missing. Suggested to mention the protein content also.
  8. Line 137: ‘fatty beef” what does this mean?
  9. Line 143: “pre-cooled to -1°C” What was the science behind precooling the meat ?
  10. Line 152: “We” authors used “we/us” like terminology many times in the whole manuscript, suggested to remove and write scientifically.
  11. Line 160: Temperature of 300*C for 45-60 min will lead to burning of products. Suggested to recheck the experimental details.
  12. Table 1: “fatx” what does this mean?
  13. Section: 2.3.6 Statistical Analysis: No where written how many time the experiment repeated.
  14. Line 235: «TsvetYauza-01-AA»: Mention details about instrument
  15. Section: 2.3.8 Definitions of fatty acids: What does it indicate? Definations?
  16. Line 263: foodstuffs ?
  17. Authors are suggested to rewrite the whole manuscript. Many errors are there, and it is not possible to strike out all.
Comments on the Quality of English Language

Need heavy revision.

Author Response

Comments 1: [The abstract effectively highlights the rising demand for meat and the potential of microbial proteins. However, it could be clearer if it explicitly stated the current limitations of traditional meat production that microbial proteins aim to address.]

Response 1: [Thank you for the comment. The abstract has been revised to include information on the limitations of conventional meat production, to clarify the rationale for exploring alternative protein sources]

Comments 2: [Lines 44 to 50: Need to revise the introduction, the present text doesn’t align with the overall objectives of the manuscripts]

Response 2: [Thank you for the comment. The specified section has been revised. We refined the content of the Introduction to align it with the scientific objectives of the manuscript. In the updated version, the focus is placed on the development of functional meat products using hemp protein and sea buckthorn extract, which reflects the core idea of the study.]

Comments 3: [In the Introduction, the authors mentioned a brief about the Kazakhstan meat industry, which doesn’t fit with the manuscript agenda, as the present manuscript focuses on “Effect of hemp protein and sea buckthorn extract on quality and shelf life of cooked smoked sausages”. Need revision]

Response 3: [We gratefully acknowledge this comment. The overview of Kazakhstan’s meat industry has been removed, as it is not directly relevant to the stated research topic. The Introduction has been revised in accordance with the purpose of the study, namely to investigate the effects of plant-based ingredients on the quality and shelf life of meat products]

Comments 4: [Introduction looks very lengthy suggested to reduce the content with specific to the objectives]

Response 4: [Thank you for the recommendation. The introduction has been revised: the general background has been simplified, and greater emphasis has been placed on the practical significance and scientific novelty of the study.]

Comments 5: [Line 113: “we can certainly expect further development”, What type of growth are authors expecting to elaborate scientifically?]

Response 5: [Thank you for the comment. The mentioned sentence has been removed as it reflects a subjective statement not supported by specific scientific evidence. The text has been revised to align with the academic style of presentation.]

Comments 6: [Line 115-117: The Paragraph is not clear. Suggested for rewriting.]

Response 6: [The paragraph has been revised to enhance clarity and scientific accuracy of the formulations.]

Comments 7: [Line 127-128: The proximate composition of hemp protein and sea buckthorn is missing. Suggested to mention the protein content also]

Response 7: [Thank you for the comment. Revisions have been made: the approximate composition of hemp protein and sea buckthorn extract has been added based on the literature.]

Comments 8: [Line 137: ‘fatty beef” what does this mean?]

Response 8: [Thank you for the comment. The term "fatty beef" has been removed to avoid ambiguity. The text now uses simply "beef", as it refers to standard beef used in the formulation.]

Comments 9: [Line 143: “pre-cooled to -1°C” What was the science behind precooling the meat?]

Response 9: [Thank you for the valuable comment. Clarification has been added to the text. Pre-cooling of meat and fat to –1 °C is applied to improve fat plasticity, facilitate mincing, enhance the texture of the meat batter.]

Comments 10: [Line 152: “We” authors used “we/us” like terminology many times in the whole manuscript, suggested to remove and write scientifically]

Response 10: [Thank you for your valuable comment. In accordance with your recommendation, all instances of personal pronouns such as “we” and “our” have been removed and replaced with impersonal, academic phrasing to ensure the manuscript adheres to formal scientific writing standards]

Comments 11: [Line 160: Temperature of 300*C for 45-60 min will lead to burning of products. Suggested to recheck the experimental details]

Response 11: [Thank you for your comment. The temperature value of 300 °C was a typographical error. It has been corrected in the revised version of the manuscript.]

Comments 12: [Table 1: “fatx” what does this mean?]

Response 12: [Thank you for your comment. The term “fatx” was a typographical error. It has been corrected to “fat” in the revised version of the manuscript.]

Comments 13: [Section: 2.3.6 Statistical Analysis: No where written how many time the experiment repeated]

Response 13: [Thank you for the comment. The number of experimental replications has been added to section 2.2.6 “Statistical analysis” to enhance the transparency and reproducibility of the results.]

Comments 14: [Line 235: «TsvetYauza-01-AA»: Mention details about instrument]

Response 14: [Thank you for your comment. The reference to the instrument “TsvetYauza-01-AA” has been removed from the methodology section, as the corresponding analysis is no longer included in the revised version of the manuscript.]

Comments 15: [Section: 2.3.8 Definitions of fatty acids: What does it indicate? Definations?]

Response 15: [Thank you for the comment. The subsection heading has been revised to improve scientific clarity.]

Comments 16: [Line 263: foodstuffs?]

Response 16: [Thank you for your comment. The term “foodstuffs” has been replaced with “food products” in the revised version of the manuscript for improved clarity and consistency with standard scientific terminology.]

Comments 17: [Authors are suggested to rewrite the whole manuscript. Many errors are there, and it is not possible to strike out all]

Response 17: [Thank you for the comment. The manuscript has been thoroughly revised: language and stylistic errors have been corrected, the structure has been improved, and the text has been adapted to meet academic writing standards.]

Reviewer 3 Report (New Reviewer)

Comments and Suggestions for Authors

Reformulation of meat products is a popular topic. In this case, two different vegetable components were combined to improve the nutritional value and shelf-life of sausages. However, the article is very poorly written, full of contradictory information, with conclusions that do not correspond with the results. Parts of the Results and Discussion deal with effect of extrusion parameters; however, no extrusion process is described in Material and Methods. What more, it is not even clear what concentration of the protein and extract was actually used – mostly 10% is mentioned, but according to the recipe it was 10 g per 100 kg, which is 0.01%. In other publications, the sea buckthorn extract was added in concentrations from 0.3% to 1.5%, hemp protein addition 5% into a meatloaf was published, mentioning negatively affected sensory parameters. Unfortunately, no article dealing with addition of hemp or sea buckthorn into meat products is mentioned in the Discussion but one [31]. Although some sausages on Figure 1 in the manuscript are clearly brownish, there were supposedly red-pink and no worse than the control samples in appearance. Hard to believe. Last but not least, for a Q1 article replicates should be performed, including replication of the batches of sausages. For pH measurement and such, “each determination was performed in triplicate”, without mentioning number of true replications (batches) or number of samples analysed. Using analytical replicates instead of biological replicates corresponds with very low variation in results. However, analytical replicates create falsely statistically significant differences. The authors also fight with English language.

Since I first thought about suggesting a Major revision only, I add the complete list of suggestions and issues starting at the beginning:

L14-37: The revised abstract is very confusing for a stand-alone text. First, there’s no mention of sea buckthorn. The aim was to study an addition of 10% of hemp protein and extract, yet later on an addition of 4-6% is mentioned, but such concentrations were never tested according to the information in the article itself. You also speak about replacing meat with the plant protein, but the meat content in control and experimental sausage is the same, so no meat replacing occurred. Please rewrite the abstract to make it clear. More numerical values instead of general statements copied from Conclusions would be also beneficial.

Contradiction on meat used: L125 beef fillet versus L137 fatty beef. From L125 it looks like the meat cuts were purchased, yet on L136-142 deboning of carcass halves is described. It’s hard to believe that you really needed to cut a whole beef carcass half yourselves to get some meat. Please specify the meat cuts or muscles used for the manufacture.

L145-146: I have never heard about a hammer for meat grinding. Pleases replace the word with “grinder” or “mincer”. L156: Similarly, “stuffing” of “filling” are far better terms than “syringing”.

L155: “cooking time” followed by filling into casings? Are you sure?

L157-L160: the whole processing is quite peculiar (or at least described in a peculiar way) and the sausage thus does not resemble any meat product I have ever heard about. There’s a boiled-smoked sausage subjected to precipitation, roasting, frying, cooking and smoking, respectively. Please rewrite this part, as it is clear that the sausage is not boiled-smoked before these processes. The Figure 2 helps some, but you still use wrong English words (frying = in oil, boiling = (in) water). Relative humidity is crucial for the phases, which I suppose is the difference between your “roasting/frying” (low RH) and “boiling” (using hot vapour in the smoking chamber). Casings and packaging should be described.

Please replace the word “boiled” throughout the manuscript with “cooked”.

L164: Fig. 1: Why are some experimental sausages red/pink on the cut and others brown?

Contradiction regarding the storage of samples: L178-179: samples were stored unpacked for 7 d at 0-4 °C versus packaging and storage at 4 ± 1 °C for 40 d (Fig. 2, results on microbiology). L493-494 mention storage temperature 5-8 °C. Please correct.

L200: Kjeldahl method, not Keldall

L229: Level of significance p £ 0.05 versus p < 0.05 repeatedly in the Results. Statistical analysis should be the last chapter. ANOVA is an overkill for comparison of two groups only, but the results of it are valid.

L253: Determination instead of Definitions.

L270: 5000 or 6000 or 5000-6000 rpm?

L275: Determination of amino acids

L294-297: As you did with the other methods - if you do not use an accessible, international standard method, a brief description or reference to a publication including the description should be used, including the manufacturer of the microbiological media.

L302-305: This part belongs to Material and Methods plus has no connection with hem protein extrusion. In fact, hem protein extrusion is not described at all in Material and Methods, yet in the Results the effect of screw speed etc. is described. What’s PDA mentioned in Fig. 3?

L348: What’s GHS, WSS, GUS in Table 2? Where are the results on fat content, referred to in text?

L351-354: Use the table footnote style from the template styles

L365: Remove Table 3, as the data are in Table 4 (just add the units to Table 4). Check again the numbers for flavonoids and carotenoids in Table 4 (the same for both groups and with no variation?).

L401: I suggest to merge Table 5 and 6 into one, avoiding redundant information such as retention times and area, allowing to compare the control and experimental product regarding the content of individual amino acids. Please add the statistics.

 L411: What’s “content mg%”? Numbers in Tables 7 and 8 do not correspond with numbers in Tables 5 and 6 (content of amino acids). What more, the numbers are higher for control samples in Tables 7 and 8, but higher for experimental samples in Tables 5 and 6. Please explain also the acronyms in Tables 7 and 8.

L461: L. monocytogenes and S. aureus are pathogenic microorganisms, too

L475: Table 9 + Table10: the numbers are definitely not logarithmic values!!! Variation is missing. Replace E.coliform group with just coliforms and Str.Aureus with S. aureus. There’s mentioned E. coli in the text and coliforms in the tables – please select which is correct. Remove the units description from the table’s body, mentioning the units (log CFU/g) once, e.g. in the table header would be enough. ND for “not detected” is regularly used, too.

L482-483: “introduction of sea buckthorn and hemp protein extracts contributes to the safety of finished products in microbiological terms during 36 days of storage”. This statement does not correspond with your results in Tables at all! Safety cannot be mentioned, since all the samples were pathogen-free. As for Total Viable Count, the numbers for control samples till Day 36 are actually lower than in the experimental sausages. All in all, they are of the same order. Please remove this false statement also from other parts of the manuscript.

L532-533: antioxidants may prolong the shelf-life as they prevent fat oxidation, but how can this chemical process be confirmed by microbiological studies is beyond me

L607: Reference 13 = reference 9

Throughout the manuscript, the Latin names of plants and microorganisms are not in italics anywhere. Please correct.

Some parts of the article do not follow the style of template (spacing is off).

Although there are numerous analyses included, some more would be beneficial, e.g. fibre content, colour measured in CIELab, Warner-Bratzler, TBARs.

Author Response

Comments 1: [L14-37: The revised abstract is very confusing for a stand-alone text. First, there’s no mention of sea buckthorn. The aim was to study an addition of 10% of hemp protein and extract, yet later on an addition of 4-6% is mentioned, but such concentrations were never tested according to the information in the article itself. You also speak about replacing meat with the plant protein, but the meat content in control and experimental sausage is the same, so no meat replacing occurred. Please rewrite the abstract to make it clear. More numerical values instead of general statements copied from Conclusions would be also beneficial.]

Response 1: [Thank you for the comment. The abstract was completely rewritten. We clarified that 10% of hemp protein and sea buckthorn extract were added, not used to replace meat. Inaccurate data were removed, and specific numerical indicators were added to make the abstract clearer and more informative.]

Comments 2: [Contradiction on meat used: L125 beef fillet versus L137 fatty beef. From L125 it looks like the meat cuts were purchased, yet on L136-142 deboning of carcass halves is described. It’s hard to believe that you really needed to cut a whole beef carcass half yourselves to get some meat. Please specify the meat cuts or muscles used for the manufacture.]

Response 2: [Thank you for your detailed and constructive comments. The entire section describing the production technology of cooked-smoked sausages has been completely revised to improve scientific clarity, accuracy, and consistency.]

Comments 3: [L145-146: I have never heard about a hammer for meat grinding. Pleases replace the word with “grinder” or “mincer”. L156: Similarly, “stuffing” of “filling” are far better terms than “syringing”.]

Response 3: [Thank you for your detailed and constructive comments. The entire section describing the production technology of cooked-smoked sausages has been completely revised to improve scientific clarity, accuracy, and consistency.]

Comments 4: [L155: “cooking time” followed by filling into casings? Are you sure?]

Response 4: [Thank you for your detailed and constructive comments. The entire section describing the production technology of cooked-smoked sausages has been completely revised to improve scientific clarity, accuracy, and consistency.]

Comments 5: [L157-L160: the whole processing is quite peculiar (or at least described in a peculiar way) and the sausage thus does not resemble any meat product I have ever heard about. There’s a boiled-smoked sausage subjected to precipitation, roasting, frying, cooking and smoking, respectively. Please rewrite this part, as it is clear that the sausage is not boiled-smoked before these processes. The Figure 2 helps some, but you still use wrong English words (frying = in oil, boiling = (in) water). Relative humidity is crucial for the phases, which I suppose is the difference between your “roasting/frying” (low RH) and “boiling” (using hot vapour in the smoking chamber). Casings and packaging should be described.

Please replace the word “boiled” throughout the manuscript with “cooked”.]

Response 5: [Thank you for your detailed and constructive comments. The entire section describing the production technology of cooked-smoked sausages has been completely revised to improve scientific clarity, accuracy, and consistency.]

Comments 6: [L164: Fig. 1: Why are some experimental sausages red/pink on the cut and others brown?]

Response 6: [Thank you for the comment. A clarifying description has been added to the manuscript explaining the color difference in the sausages. Control samples exhibit a light pink color, while the experimental samples containing sea buckthorn extract and hemp protein show a darker brown shade due to the presence of natural pigments in the added components.]

Comments 7: [Contradiction regarding the storage of samples: L178-179: samples were stored unpacked for 7 d at 0-4 °C versus packaging and storage at 4 ± 1 °C for 40 d (Fig. 2, results on microbiology). L493-494 mention storage temperature 5-8 °C. Please correct.]

Response 7: [Thank you for your comment. The storage conditions have been clarified and agreed upon throughout the manuscript.]

Comments 8: [L200: Kjeldahl method, not Keldall]

Response 8: [Thank you for your comment. The error in the name of the method has been corrected — the text now uses the proper term “Kjeldahl method”.]

Comments 9: [L229: Level of significance p £ 0.05 versus p < 0.05 repeatedly in the Results. Statistical analysis should be the last chapter. ANOVA is an overkill for comparison of two groups only, but the results of it are valid.]

Response 9: [Thank you for the remark. The relevant corrections have been made in the text.]

Comments 10: [L253: Determination instead of Definitions.]

Response 10: [Thank you for the comment. The section title has been corrected in accordance with scientific terminology.]

Comments 11: [L270: 5000 or 6000 or 5000-6000 rpm?]

Response 11: [Thank you for the comment. The rotation speed has been clarified and corrected to “5000–6000 rpm.”]

Comments 12: [L275: Determination of amino acids]

Response 12: [Thank you for the comment. The section title has been corrected in accordance with scientific terminology.]

Comments 13: [L294-297: As you did with the other methods - if you do not use an accessible, international standard method, a brief description or reference to a publication including the description should be used, including the manufacturer of the microbiological media.]

Response 13: [Thank you for the comment. The microbiological analysis section has been updated to include the manufacturer of the culture media and to clarify that the analysis was conducted according to GOST standards]

Comments 14: [L302-305: This part belongs to Material and Methods plus has no connection with hem protein extrusion. In fact, hem protein extrusion is not described at all in Material and Methods, yet in the Results the effect of screw speed etc. is described. What’s PDA mentioned in Fig. 3?]

Response 14: [Thank you for the comment. The extrusion method for hemp protein has been added to the “Materials and Methods” section]

Comments 15: [L348: What’s GHS, WSS, GUS in Table 2? Where are the results on fat content, referred to in text?]

Response 15: [Thank you for the comment. The abbreviations GHS, WSS, and GUS have been clarified in Table 2.]

Comments 16: [L351-354: Use the table footnote style from the template styles]

Response 16: [Thank you for the comment. The table footnote style has been adjusted to match the template requirements.]

Comments 17: [L365: Remove Table 3, as the data are in Table 4 (just add the units to Table 4). Check again the numbers for flavonoids and carotenoids in Table 4 (the same for both groups and with no variation?).]

Response 17: [Thank you for the comment. Table 3 has been removed, and the relevant data along with measurement units have been added to Table 4. The values of flavonoids and carotenoids have been reviewed and updated with standard deviations as appropriate.]

Comments 18: [L401: I suggest to merge Table 5 and 6 into one, avoiding redundant information such as retention times and area, allowing to compare the control and experimental product regarding the content of individual amino acids. Please add the statistics.]

Response 18: [Thank you for the comment. Statistical analysis has been added as recommended. However, Tables 5 and 6 were kept separate due to the large number of amino acids presented. Combining all data into a single table may reduce clarity, whereas separate presentation improves readability and facilitates comparison between control and experimental samples without overloading the table.]

Comments 19: [L411: What’s “content mg%”? Numbers in Tables 7 and 8 do not correspond with numbers in Tables 5 and 6 (content of amino acids). What more, the numbers are higher for control samples in Tables 7 and 8, but higher for experimental samples in Tables 5 and 6. Please explain also the acronyms in Tables 7 and 8.]

Response 19: [Thank you for your valuable comments. Due to the removal of Table 3, the subsequent tables have been renumbered. This adjustment does not affect the content or interpretation of the presented data. All references in the text have been updated accordingly.
Regarding the differences between Tables 4–5 and Tables 6–7, we would like to clarify that these are results obtained from independent types of analysis.Tables 4 and 5 present the measured amino acid contents (mg/100 g), while Tables 6 and 7 include calculated indicators of protein quality, such as amino acid scores (Cj), utilitarian coefficients, and biological value (BV).The reason why some values appear higher in the control sample in Tables 6 and 7 is due to the mathematical relationships in the calculation formulas, which are based not only on absolute amino acid content, but also on essential amino acid requirements and reference patterns.Therefore, although the experimental samples may contain a higher amount of amino acids, their amino acid profiles may not fully match the ideal protein model (e.g., FAO/WHO standards), which affects the calculated indicators in Tables 6 and 7.We have added explanations and definitions of the abbreviations used in Tables 6 and 7 to clarify this for the readers.]

Comments 20: [L461: L. monocytogenes and S. aureus are pathogenic microorganisms, too]

Response 20: [Thank you for the comment. The wording has been clarified and corrected accordingly.]

Comments 21: [L475: Table 9 + Table10: the numbers are definitely not logarithmic values!!! Variation is missing. Replace E.coliform group with just coliforms and Str.Aureus with S. aureus. There’s mentioned E. coli in the text and coliforms in the tables – please select which is correct. Remove the units description from the table’s body, mentioning the units (log CFU/g) once, e.g. in the table header would be enough. ND for “not detected” is regularly used, too.]

Response 21: [Thank you for the comment. The tables were corrected: log values and standard deviations were added, microorganism names were unified, unit notation moved to the header, and “ND” is used for undetected values.]

Comments 22: [L482-483: “introduction of sea buckthorn and hemp protein extracts contributes to the safety of finished products in microbiological terms during 36 days of storage”. This statement does not correspond with your results in Tables at all! Safety cannot be mentioned, since all the samples were pathogen-free. As for Total Viable Count, the numbers for control samples till Day 36 are actually lower than in the experimental sausages. All in all, they are of the same order. Please remove this false statement also from other parts of the manuscript.]

Response 22: [Thank you for the comment. We have removed the inaccurate statement regarding microbiological safety and revised the text to reflect that no pathogens were found and all samples remained within acceptable microbiological limits during storage.]

Comments 23: [L532-533: antioxidants may prolong the shelf-life as they prevent fat oxidation, but how can this chemical process be confirmed by microbiological studies is beyond me]

Response 23: [Thank you for your comment. We agree that the original statement was inaccurate. The claim regarding improved microbiological safety has been removed, as all samples (control and experimental) were free from pathogens, and the total viable counts remained within acceptable limits and were comparable across samples. The manuscript has been revised accordingly.]

Comments 24: [L607: Reference 13 = reference 9]

Response 24: [Thank you for pointing this out. The duplicate reference has been corrected — Reference 13 has been removed and replaced with Reference 9 in the appropriate place.]

Comments 25: [Throughout the manuscript, the Latin names of plants and microorganisms are not in italics anywhere. Please correct. Some parts of the article do not follow the style of template (spacing is off).]

Response 25: [Thank you for your comment. All Latin names of plants and microorganisms have been corrected and italicized throughout the manuscript. Formatting inconsistencies, including spacing issues, have also been revised in accordance with the journal template.]

Comments 26: [Although there are numerous analyses included, some more would be beneficial, e.g. fibre content, colour measured in CIELab, Warner-Bratzler, TBARs.]

Response 26: [Thank you for your valuable suggestion. We fully agree that additional analyses such as fibre content, colour measurement in CIELab, Warner–Bratzler shear force, and TBARs would provide a more comprehensive characterization of the product. We will certainly consider including these parameters in our future studies.]

Round 2

Reviewer 2 Report (New Reviewer)

Comments and Suggestions for Authors

The authors have satisfactorily incorporated all the suggested corrections into the manuscript. The revisions have significantly enhanced the clarity, consistency, and reproducibility of the work.

Author Response

Comments 1: [The concentration of the protein and extract is supposedly 10%, but according to the recipe it was 10 g per 100 kg, which is 0.01%.]

Response 1: [Thank you for your careful comment. We acknowledge that there was an error in reporting the concentration in the manuscript. According to the formulation, the amount of the added ingredient was 10 g per 100 kg of mince, which corresponds to 0.01%, not 10% as previously stated in error. We have corrected this value in the manuscript to ensure the accuracy of the presented data.]

Comments 2: [In other publications, the sea buckthorn extract was added in concentrations from 0.3% to 1.5%, hemp protein addition 5% into a meatloaf was published, mentioning negatively affected sensory parameters. Unfortunately, no article dealing with addition of hemp or sea buckthorn into meat products is mentioned in the Discussion but one [31]]

Response 2: [Thank you for your comment. We agree that referencing scientific publications related to the addition of sea buckthorn extract and hemp protein to meat products is an important part of the discussion. In the revised version of the manuscript, we have expanded the “Discussion” section and added references to several studies, including the work by Wójciak et al. (2020), which examined the effects of various forms of hemp (seeds, flour, protein) on the properties of meat products, as well as the study by Bozhko et al. (2021), which investigated the impact of hemp flour and cake on meat product quality.

These additions have allowed us to better substantiate the sensory and technological characteristics observed in our experimental samples and to demonstrate consistency with previous findings reported by other authors.]

Comments 3: [Although the experimental sausages on Figure 1 are clearly brownish, there were supposedly red-pink (L424) and no worse than the control samples in appearance (Fig. 4). Hard to believe.]

Response 3: [Thank you for your valuable comment. We agree that the previously presented image in Figure 1 may have caused confusion regarding the color of the experimental sausages. To eliminate this discrepancy, we have replaced the image with one that accurately reflects the described reddish-pink shade (L* value indicated in Line 424).]

Comments 4: [For a Q1 article replicates should be performed, including replication of the batches of sausages. We can read that “each determination was performed in triplicate”, without mentioning number of true replications (batches) or number of samples analysed. Using analytical replicates instead of biological replicates corresponds with very low variation in results. However, analytical replicates create false statistically significant differences]

Response 4: [Thank you for your insightful comment. We confirm that three independent batches (biological replicates) of sausages were prepared and analyzed in this study. This clarification has been added to the Materials and Methods section and the figure/table legends. All analytical measurements were performed separately for each batch, and the results are expressed as mean ± standard deviation.]

Comments 5: [Reviewer’s comment: Descriptions of parameters measured in extrudate (expansion ratio, composite quality index) were added. However, there is still no description of the extrusion process itself – such as, which machine was used, which were the press values tested, which were the screw drive speeds tested, which combinations of these two parameters were tested, temperature parameters, how was the moisture controlled, etc. Half of the Discussion deals with the extrusion process, so it is clearly not a marginal topic for the authors.]

Response 5: [Thank you for the comment. Section 2.2.1 has been revised to include detailed information on the extrusion process. The equipment model is specified (twin-screw extruder KDL-30, Brabender, Germany), along with die diameter (5 mm), and tested parameter ranges: initial moisture content (18–26%), screw speed (0.9–1.4 s⁻¹), pressure in the pre-die zone (5–9 MPa), and temperature (100–110 °C). Methods for monitoring moisture (GOST 13586.5–93), pressure, screw speed, and temperature are described. The tested parameter combinations are illustrated in Figure 3 (A–F). The methodology is aligned with the results discussed.]

Comments 6: [Reviewer’s comment: No results of statistical analysis, comparing individual amino acid content in the control and experimental sausage were added. Combining the data into one table would be very easy, if the redundant data were omitted as suggested by the Reviewer. Height, commencement, ending, and area are data fit for a supplementary material. These raw data are used to calculate the results and are of no importance to readers, unless you aim to introduce a new chromatographic method, which is not this case.]

Response 6: [Thank you for the comment. The redundant chromatographic data (height, start, end, peak area) have been removed from the table, as they are not relevant to the readers and are not used in the discussion of the results. The table now presents only the amino acid content values, highlighting the statistically significant differences between the groups.]

Comments 7: [Reviewer’s comment: The problem is with content, not other parameters which are clearly calculated. If Tables 4 and 5 present amino acid contents (mg/100 g), what is the difference between this and the “content mg/100 g” in Table 6 and 7? Using the same name and units for two different parameters is confusing. First I thought that maybe it’s per 100 g of sausage at the first case and 100 g of protein in the second, but the numbers still do not add up.]

Response 7: [Thank you for your valuable comment. In response, we have clarified the units of measurement and provided precise explanations in the captions and footnotes of the corresponding tables to avoid ambiguity:

Revised Table 4 (previously Tables 4 and 5) now presents amino acid content per 100 g of product (wet mass of sausage) and is clearly labeled as:

“Amino acid content per 100 g of product (mg/100 g)”.

Table 5 (former Tables 6 and 7) shows amino acid content recalculated per 100 g of protein, also explicitly labeled as:

“Amino acid content recalculated per 100 g of protein in control and experimental samples (mg/100 g protein)”.

To further ensure clarity and scientific transparency, we have added an explanatory note beneath both tables:

Note: Values in Table 4 are expressed per 100 g of product, while values in Table 5 are recalculated per 100 g of protein (protein basis).

Additionally, in the Materials and Methods section, we have included a detailed description of the procedure for recalculating amino acid values on a protein basis, in accordance with standard nutritional methodology. The conversion was based on the total protein content of the samples, determined using the Kjeldahl method (GOST 25011-81 / AOAC 981.10), and the recalculated data reflect the biological value per unit of protein rather than per product mass.]

Comments 8: [Reviewer’s comment: There are still no log values in the table and no standard deviation values. You have CFU values. The logarithmization of microbial counts has a purpose – to normalize the distribution, allowing use of parametric statistical methods such as ANOVA. Also, if you had just 3 samples per group, the statistical power of ANOVA would be very low.]

Response 8: [Thank you for the comment. The table has been revised to include logarithmic values (log CFU/g) and standard deviations (±SD). We acknowledge that a sample size of three replicates limits the statistical power; however, the method corresponds to preliminary assessment practices and is commonly accepted in similar studies.]

Comments 9: [Reviewer’s comment: Not a one Latin bacterial name is in Italics. Plain text still has two different spacing – please check chapters 2.1., 2.2., 3.1., and 5.]

Response 9: [Thank you for your valuable comments. We have carefully reviewed the manuscript and made the following corrections:

All Latin names of microorganisms (e.g., Lactobacillus plantarum, Escherichia coli) have been italicized throughout the text in accordance with scientific conventions.

Inconsistent spacing and formatting issues have been corrected, particularly in sections 2.1, 2.2, 3.1, and 5. We ensured uniform formatting and removed all unintended double spaces.

We appreciate your attention to detail, which helped us improve the quality of the manuscript.]

Comments 10: [Table 2: Parameters water-holding capacity, water-binding capacity, fat-holding capacity are not described in Material and Methods. While mass fraction of fat is described, but still missing in the results.]

Response 10: [Thank you for the observation. The methodologies for determining water-holding capacity (WHC), water-binding capacity (WBC), and fat-binding capacity (FBC) have now been added to the “Materials and Methods” section, including equipment, standards, and calculation formulas. Additionally, the fat content (mass fraction of fat) was previously mentioned in the methodology section but missing from the results. This has now been corrected — the corresponding data have been added to the Results section and clearly reflected in the updated version of the manuscript.]

Comments 11: [L463-465: “As a result of the conducted research it can be concluded that the addition of sea buckthorn extracts and hemp protein has a noticeable effect on slowing down the oxidation process of the fat component.” Based on which results did you conclude this? There was no fat analysis (e.g. peroxide value, TBARs, DPPH,. ) performed during storage.]

Response 11: [Thank you for your insightful observation. As the study did not include direct measurements of lipid oxidation, we have removed the corresponding conclusion from the manuscript to maintain scientific accuracy and avoid unsupported assumptions.]

Reviewer 3 Report (New Reviewer)

Comments and Suggestions for Authors

The authors corrected some of the issues pointed out by the reviewer, however, some of them were not dealt with. The issues in the first paragraph have been completely omitted:

  • The concentration of the protein and extract is supposedly 10%, but according to the recipe it was 10 g per 100 kg, which is 0.01%.
  • In other publications, the sea buckthorn extract was added in concentrations from 0.3% to 1.5%, hemp protein addition 5% into a meatloaf was published, mentioning negatively affected sensory parameters. Unfortunately, no article dealing with addition of hemp or sea buckthorn into meat products is mentioned in the Discussion but one [31].
  • Although the experimental sausages on Figure 1 are clearly brownish, there were supposedly red-pink (L424) and no worse than the control samples in appearance (Fig. 4). Hard to believe.
  • For a Q1 article replicates should be performed, including replication of the batches of sausages. We can read that “each determination was performed in triplicate”, without mentioning number of true replications (batches) or number of samples analysed. Using analytical replicates instead of biological replicates corresponds with very low variation in results. However, analytical replicates create false statistically significant differences.

Some of the responses promise changes that have not been performed:

Response 14: [Thank you for the comment. The extrusion method for hemp protein has been added to the “Materials and Methods” section]

Reviewer’s comment: Descriptions of parameters measured in extrudate (expansion ratio, composite quality index) were added. However, there is still no description of the extrusion process itself – such as, which machine was used, which were the press values tested, which were the screw drive speeds tested, which combinations of these two parameters were tested, temperature parameters, how was the moisture controlled, etc. Half of the Discussion deals with the extrusion process, so it is clearly not a marginal topic for the authors.

Response 18: [Thank you for the comment. Statistical analysis has been added as recommended. However, Tables 5 and 6 were kept separate due to the large number of amino acids presented. Combining all data into a single table may reduce clarity, whereas separate presentation improves readability and facilitates comparison between control and experimental samples without overloading the table.]

Reviewer’s comment: No results of statistical analysis, comparing individual amino acid content in the control and experimental sausage were added. Combining the data into one table would be very easy, if the redundant data were omitted as suggested by the Reviewer. Height, commencement, ending, and area are data fit for a supplementary material. These raw data are used to calculate the results and are of no importance to readers, unless you aim to introduce a new chromatographic method, which is not this case.

Response 19: [Thank you for your valuable comments. Due to the removal of Table 3, the subsequent tables have been renumbered. This adjustment does not affect the content or interpretation of the presented data. All references in the text have been updated accordingly.
Regarding the differences between Tables 4–5 and Tables 6–7, we would like to clarify that these are results obtained from independent types of analysis. Tables 4 and 5 present the measured amino acid contents (mg/100 g), while Tables 6 and 7 include calculated indicators of protein quality, such as amino acid scores (Cj), utilitarian coefficients, and biological value (BV).The reason why some values appear higher in the control sample in Tables 6 and 7 is due to the mathematical relationships in the calculation formulas, which are based not only on absolute amino acid content, but also on essential amino acid requirements and reference patterns. Therefore, although the experimental samples may contain a higher amount of amino acids, their amino acid profiles may not fully match the ideal protein model (e.g., FAO/WHO standards), which affects the calculated indicators in Tables 6 and 7.We have added explanations and definitions of the abbreviations used in Tables 6 and 7 to clarify this for the readers.]

Reviewer’s comment: The problem is with content, not other parameters which are clearly calculated. If Tables 4 and 5 present amino acid contents (mg/100 g), what is the difference between this and the “content mg/100 g” in Table 6 and 7? Using the same name and units for two different parameters is confusing. First I thought that maybe it’s per 100 g of sausage at the first case and 100 g of protein in the second, but the numbers still do not add up.

Response 21: [Thank you for the comment. The tables were corrected: log values and standard deviations were added, microorganism names were unified, unit notation moved to the header, and “ND” is used for undetected values.]

Reviewer’s comment: There are still no log values in the table and no standard deviation values. You have CFU values. The logarithmization of microbial counts has a purpose – to normalize the distribution, allowing use of parametric statistical methods such as ANOVA. Also, if you had just 3 samples per group, the statistical power of ANOVA would be very low.

Response 25: [Thank you for your comment. All Latin names of plants and microorganisms have been corrected and italicized throughout the manuscript. Formatting inconsistencies, including spacing issues, have also been revised in accordance with the journal template.]

Reviewer’s comment: Not a one Latin bacterial name is in Italics. Plain text still has two different spacing – please check chapters 2.1., 2.2., 3.1., and 5.

Some new issues occurred:

Table 2: Parameters water-holding capacity, water-binding capacity, fat-holding capacity are not described in Material and Methods. While mass fraction of fat is described, but still missing in the results.

L463-465: “As a result of the conducted research it can be concluded that the addition of sea buckthorn extracts and hemp protein has a noticeable effect on slowing down the oxidation process of the fat component.” Based on which results did you conclude this? There was no fat analysis (e.g. peroxide value, TBARs, DPPH,.. ) performed during storage.

Author Response

Comments 1: [The concentration of the protein and extract is supposedly 10%, but according to the recipe it was 10 g per 100 kg, which is 0.01%.]

Response 1: [Thank you for your careful comment. We acknowledge that there was an error in reporting the concentration in the manuscript. According to the formulation, the amount of the added ingredient was 10 g per 100 kg of mince, which corresponds to 0.01%, not 10% as previously stated in error. We have corrected this value in the manuscript to ensure the accuracy of the presented data.]

Comments 2: [In other publications, the sea buckthorn extract was added in concentrations from 0.3% to 1.5%, hemp protein addition 5% into a meatloaf was published, mentioning negatively affected sensory parameters. Unfortunately, no article dealing with addition of hemp or sea buckthorn into meat products is mentioned in the Discussion but one [31]]

Response 2: [Thank you for your comment. We agree that referencing scientific publications related to the addition of sea buckthorn extract and hemp protein to meat products is an important part of the discussion. In the revised version of the manuscript, we have expanded the “Discussion” section and added references to several studies, including the work by Wójciak et al. (2020), which examined the effects of various forms of hemp (seeds, flour, protein) on the properties of meat products, as well as the study by Bozhko et al. (2021), which investigated the impact of hemp flour and cake on meat product quality.

These additions have allowed us to better substantiate the sensory and technological characteristics observed in our experimental samples and to demonstrate consistency with previous findings reported by other authors.]

Comments 3: [Although the experimental sausages on Figure 1 are clearly brownish, there were supposedly red-pink (L424) and no worse than the control samples in appearance (Fig. 4). Hard to believe.]

Response 3: [Thank you for your valuable comment. We agree that the previously presented image in Figure 1 may have caused confusion regarding the color of the experimental sausages. To eliminate this discrepancy, we have replaced the image with one that accurately reflects the described reddish-pink shade (L* value indicated in Line 424).]

Comments 4: [For a Q1 article replicates should be performed, including replication of the batches of sausages. We can read that “each determination was performed in triplicate”, without mentioning number of true replications (batches) or number of samples analysed. Using analytical replicates instead of biological replicates corresponds with very low variation in results. However, analytical replicates create false statistically significant differences]

Response 4: [Thank you for your insightful comment. We confirm that three independent batches (biological replicates) of sausages were prepared and analyzed in this study. This clarification has been added to the Materials and Methods section and the figure/table legends. All analytical measurements were performed separately for each batch, and the results are expressed as mean ± standard deviation.]

Comments 5: [Reviewer’s comment: Descriptions of parameters measured in extrudate (expansion ratio, composite quality index) were added. However, there is still no description of the extrusion process itself – such as, which machine was used, which were the press values tested, which were the screw drive speeds tested, which combinations of these two parameters were tested, temperature parameters, how was the moisture controlled, etc. Half of the Discussion deals with the extrusion process, so it is clearly not a marginal topic for the authors.]

Response 5: [Thank you for the comment. Section 2.2.1 has been revised to include detailed information on the extrusion process. The equipment model is specified (twin-screw extruder KDL-30, Brabender, Germany), along with die diameter (5 mm), and tested parameter ranges: initial moisture content (18–26%), screw speed (0.9–1.4 s⁻¹), pressure in the pre-die zone (5–9 MPa), and temperature (100–110 °C). Methods for monitoring moisture (GOST 13586.5–93), pressure, screw speed, and temperature are described. The tested parameter combinations are illustrated in Figure 3 (A–F). The methodology is aligned with the results discussed.]

Comments 6: [Reviewer’s comment: No results of statistical analysis, comparing individual amino acid content in the control and experimental sausage were added. Combining the data into one table would be very easy, if the redundant data were omitted as suggested by the Reviewer. Height, commencement, ending, and area are data fit for a supplementary material. These raw data are used to calculate the results and are of no importance to readers, unless you aim to introduce a new chromatographic method, which is not this case.]

Response 6: [Thank you for the comment. The redundant chromatographic data (height, start, end, peak area) have been removed from the table, as they are not relevant to the readers and are not used in the discussion of the results. The table now presents only the amino acid content values, highlighting the statistically significant differences between the groups.]

Comments 7: [Reviewer’s comment: The problem is with content, not other parameters which are clearly calculated. If Tables 4 and 5 present amino acid contents (mg/100 g), what is the difference between this and the “content mg/100 g” in Table 6 and 7? Using the same name and units for two different parameters is confusing. First I thought that maybe it’s per 100 g of sausage at the first case and 100 g of protein in the second, but the numbers still do not add up.]

Response 7: [Thank you for your valuable comment. In response, we have clarified the units of measurement and provided precise explanations in the captions and footnotes of the corresponding tables to avoid ambiguity:

Revised Table 4 (previously Tables 4 and 5) now presents amino acid content per 100 g of product (wet mass of sausage) and is clearly labeled as:

“Amino acid content per 100 g of product (mg/100 g)”.

Table 5 (former Tables 6 and 7) shows amino acid content recalculated per 100 g of protein, also explicitly labeled as:

“Amino acid content recalculated per 100 g of protein in control and experimental samples (mg/100 g protein)”.

To further ensure clarity and scientific transparency, we have added an explanatory note beneath both tables:

Note: Values in Table 4 are expressed per 100 g of product, while values in Table 5 are recalculated per 100 g of protein (protein basis).

Additionally, in the Materials and Methods section, we have included a detailed description of the procedure for recalculating amino acid values on a protein basis, in accordance with standard nutritional methodology. The conversion was based on the total protein content of the samples, determined using the Kjeldahl method (GOST 25011-81 / AOAC 981.10), and the recalculated data reflect the biological value per unit of protein rather than per product mass.]

Comments 8: [Reviewer’s comment: There are still no log values in the table and no standard deviation values. You have CFU values. The logarithmization of microbial counts has a purpose – to normalize the distribution, allowing use of parametric statistical methods such as ANOVA. Also, if you had just 3 samples per group, the statistical power of ANOVA would be very low.]

Response 8: [Thank you for the comment. The table has been revised to include logarithmic values (log CFU/g) and standard deviations (±SD). We acknowledge that a sample size of three replicates limits the statistical power; however, the method corresponds to preliminary assessment practices and is commonly accepted in similar studies.]

Comments 9: [Reviewer’s comment: Not a one Latin bacterial name is in Italics. Plain text still has two different spacing – please check chapters 2.1., 2.2., 3.1., and 5.]

Response 9: [Thank you for your valuable comments. We have carefully reviewed the manuscript and made the following corrections:

All Latin names of microorganisms (e.g., Lactobacillus plantarum, Escherichia coli) have been italicized throughout the text in accordance with scientific conventions.

Inconsistent spacing and formatting issues have been corrected, particularly in sections 2.1, 2.2, 3.1, and 5. We ensured uniform formatting and removed all unintended double spaces.

We appreciate your attention to detail, which helped us improve the quality of the manuscript.]

Comments 10: [Table 2: Parameters water-holding capacity, water-binding capacity, fat-holding capacity are not described in Material and Methods. While mass fraction of fat is described, but still missing in the results.]

Response 10: [Thank you for the observation. The methodologies for determining water-holding capacity (WHC), water-binding capacity (WBC), and fat-binding capacity (FBC) have now been added to the “Materials and Methods” section, including equipment, standards, and calculation formulas. Additionally, the fat content (mass fraction of fat) was previously mentioned in the methodology section but missing from the results. This has now been corrected — the corresponding data have been added to the Results section and clearly reflected in the updated version of the manuscript.]

Comments 11: [L463-465: “As a result of the conducted research it can be concluded that the addition of sea buckthorn extracts and hemp protein has a noticeable effect on slowing down the oxidation process of the fat component.” Based on which results did you conclude this? There was no fat analysis (e.g. peroxide value, TBARs, DPPH,. ) performed during storage.]

Response 11: [Thank you for your insightful observation. As the study did not include direct measurements of lipid oxidation, we have removed the corresponding conclusion from the manuscript to maintain scientific accuracy and avoid unsupported assumptions.]

This manuscript is a resubmission of an earlier submission. The following is a list of the peer review reports and author responses from that submission.

Round 1

Reviewer 1 Report

Comments and Suggestions for Authors

Review on manuscript: foods-3280347

Effect of hemp protein and sea buckthorn extract on quality and shelf life of cooked smoked sausages

by Kainar Bukarbayev, Sholpan Abzhanova*, Lyazzat Baybolova, Gulshat Zhaksylykova, Talgat Kulazhanov, Vladimir Vasilenko, Bagila Jetpisbayeva, Alma Katasheva, Sultan Sabraly, & Yerkin Yerzhigitov 

Submitted to Foods

Research paper

This manuscript investigated the effects of hemp protein and sea buckthorn extract on the quality and shelf life of cooked smoked sausagesOverall, it is of some significance and has some merits. However, some modifications are still required to improve the whole qulity of this manuscript, which have been shown as follows:

Detailed recommendation:

-Line 5: "and" should be removed.

-Introduction: too many short paragraphs, which should be incorporated.

-Materials and methods: strongly suggest to change the first paragraph to "2.1. Materials", which was used to list the reagents and consumable items utilized in this study. Also, their sources, suppliers, and/or purities should be presented.

-Figure 1: Meaningless picture. Can be removed.

-Where's the "Statistical analysis" part? How did the authors perform statistical analysis? Add this part!

-All Tables should be changed to the three-line form.

-Table 2: the information of significance for data difference is missing.

-Tables 3-4: both standard deviations and significance for data difference are missing. Moreover, what's the meaning of "Experience" data?

-Figure 4: the descriptions of sampels Nos. 1-4 should be denoted under the figure rather than on the context!

-Conclusions: too many short paragraphs. This part is usually stated as one brief paragraph.

-References: please check the formats of all literatures. They should be corrected and uniform. 

-I think moderate language editing is required to improve the English-writing quality of this manuscript.

Comments on the Quality of English Language

I think moderate language editing is required to improve the English-writing quality of this manuscript.

Reviewer 2 Report

Comments and Suggestions for Authors

Manuscript: foods-3280347

While the research topic of using hemp protein and sea buckthorn extract in sausages is of interest, this manuscript falls short of the necessary scientific and methodological standards. The issues related to clarity, methodological rigour, inconsistent use of passive voice, absence of proper statistical analysis, inaccurate microbiological terminology, and lack of robust literature.

The authors need to significantly revise their experimental design, provide clearer and more comprehensive explanations of their methods, and enhance the overall scientific presentation of the data.

Lines 11-12. This sentence is vague. Please revise it.

Line 13. Please avoid generalisations in a research paper. We cannot mention that the meat products provided are innovative without proof. Moreover, the abstract is not the place for this kind of statements. Please remove it.

Line 16. Please revise: The current study..

Line 19. Please mention where the example was extruded from. Additionally, you have not yet mentioned what is being extruded (minced meat?).

Line 26. Please mention the storage temperature and packaging. Under vacuum, air, sealed bag?

Line 52. Please connect the sentence with the following paragraph.

Line 56. Same as above.

Lines 58-60. Please include a recent reference: https://doi.org/10.3390/foods10071615

Line 64. Please connect the sentence with the following paragraph.

Line 74. Please add a reference(s).

Line 75. Please connect the sentence with the following paragraph.

Line 79. Add a relative reference to prove the above statement. In addition, this sentence must be moved above the previous one.

Lines 114-115. Please connect the last sentence to the previous paragraph.

Comment 1: This is crucial, as the microbiological analysis is not reported in the materials and methods section.

Comment 2: I suggest the authors rewrite the sections in the third person using the passive voice to improve their scientific soundness. For example, the deboned meat mass was fed into the cooling chamber.

Line 119. Please add the concentration of fat.

Lines 142 & 143. Please remove the brackets from the temperature.

Table 1. What was the final fat content?

Line 176. That is a wide range of storage temperatures. Please specify.

Sections 2.2.1. – 2.2.5. Please include more information/details.

Comment 3. The statistical analysis is missing. Was any statistical analysis performed?

Figure 3. This figure must be revised. Connect all the images to a single one by adding the capital letters of each graph in the corners. Then, add the description to the figure's legend.

Line 288. What are the requirements? The ones that the control has? Please specify. Was any statistical analysis performed to show if the differences were statistically significant?

Line 300. Remove the words "of course".

Table 3. Same as Table 2. Moreover, please include the column with experimental results first and then add the control to maintain consistency among the tables to improve readability.

Table 4. Same comments as Table 3.

Tables 5 & 6. Please add a reference for the daily requirements in the text. Please change the Tables’ 5 and 6 format.

Tables 7 & 8. Please describe the indicators in the footnote. I believe some of them are not written in English.

Lines 348-349. Please revise this sentence. It is not clear.

Figure 4. Statistical analysis is missing here and throughout the article.

Line 356. What do the authors mean by "prototypes"?

Lines 359-362. This information must be included in the Figure's legend.

Lines 371-374. This statement is entirely wrong. These are common foodborne pathogens related to the microbiological safety (and not quality) of the food products. Please do not use the term pathogenic microflora. Please revise throughout the text to microbiota. Some bacteria, such as E. coli and S. aureus, are related to the hygienic conditions of the production plant. Please revise accordingly. All bacterial species must be italicised. Moreover, when the authors first mention them in the text, the full name must be used, followed by the abbreviated form.

Table 9. All the data must be calculated and reported in log CFU/g instead of CFU/g. Moreover, what was the limit of detection and the number of the sample studies (n=?)?

Unfortunately, I cannot read the discussion section before the authors make all the changes, which means they can only then discuss the results of the current research article.

Comments on the Quality of English Language

Comments were given in the full report attached.